



# The unidentified volcanic eruption of 1809: why it remains a climatic cold case

Claudia Timmreck[1], Matthew Toohey[2], Davide Zanchettin[3], Stefan Brönnimann[4], Elin Lundstad[4], and Rob Wilson[5]

[1]The Atmosphere in the Earth System, Max Planck Institute for Meteorology, Bundesstr. 53, 20146 Hamburg, Germany
[2]Department of Physics and Engineering Physics, University of Saskatchewan, Saskatoon, Canada
[3]Department of Environmental Sciences, Informatics and Statistics, University Ca' Foscari of Venice, Mestre, Italy
[4]Institute of Geography Climatology and Oeschger Centre for Climate Change Research, University of Bern, 3012 Bern, Switzerland
[5]School of Earth & Environmental Sciences, University of St. Andrews, United Kingdom

**Correspondence:** Claudia Timmreck (claudia.timmreck@mpimet.mpg.de)

**Abstract.** The "1809 eruption" is one of the most recent unidentified volcanic eruptions with a global climate impact. Even though the eruption ranks as the 3rd largest since 1500 with an eruption magnitude estimated to be two times that of the 1991 eruption of Pinatubo, not much is known of it from historic sources. Based on a compilation of instrumental and reconstructed temperature time series, we show here that tropical temperatures show a significant drop in response to the ~1809 eruption, similar to that produced by the Mt. Tambora eruption in 1815, while the response of Northern Hemisphere (NH) boreal summer temperature is spatially heterogeneous. We test the sensitivity of the climate response simulated by the MPI Earth system model to a range of volcanic forcing estimates constructed using estimated volcanic stratospheric sulfur injections (VSSI) and uncertainties from ice core records. Three of the forcing reconstructions represent a tropical eruption with approximately symmetric hemispheric aerosol spread but different forcing magnitudes, while a fourth reflects a hemispherically asymmetric scenario without volcanic forcing in the NH extratropics. Observed and reconstructed post-volcanic surface NH summer temperature anomalies lie within the range of all the scenario simulations. Therefore, assuming the model climate sensitivity is correct, the VSSI estimate is accurate within the uncertainty bounds. Comparison of observed and simulated tropical temperature anomalies suggests that the most likely VSSI for the 1809 eruption would be somewhere between 12 -19 Tg of sulfur. Model results show that NH large-scale climate modes are sensitive to both volcanic forcing strength and its spatial structure. While spatial correlations between the N-TREND NH temperature reconstruction and the model simulations are weak in terms of the ensemble mean model results, individual model simulations show good correlation over North America and Europe, suggesting the spatial heterogeneity of the 1810 cooling could be due to internal climate variability.

## 1 Introduction

The early 19th century (~1800-1830 CE), at the tail end of the Little Ice Age, marks the coldest period of the past 500 years and is therefore of special interest in the study of inter-decadal climate variability (Jungclaus et al., 2017). It was influenced by strong natural forcing: a grand solar minimum (Dalton Minimum, ~1790-1820 CE) and simultaneously a cluster of very strong





tropical volcanic eruptions that includes the widely known Mt. Tambora eruption in 1815, an unidentified eruption estimated to have occurred in 1808 or 1809, and a series of eruptions in the 1820s and 1830s. Brönnimann et al. (2019a) point out that this sequence of volcanic eruptions influenced the last phase of the Little Ice Age by not only leading to global cooling but also by
modifying the large-scale atmospheric circulation through a southward shift of low-pressure systems over the North Atlantic related to a weakening of the African monsoon and the Atlantic–European Hadley cell (Wegmann et al., 2014).

    The Mt. Tambora eruption in April 1815 was the largest in the last 500 years and had substantial global climatic and societal effects (e.g., Oppenheimer, 2003; Brönnimann and Krämer, 2016; Raible et al., 2016). In contrast to the Mt. Tambora eruption, little is known about the 1809 eruption. Although there is no historical source reporting a strong volcanic eruption
in 1809, its occurrence is indubitably brought to light by ice core sulfur records, which clearly identify a peak in volcanic sulfur in 1809/1810 (Dai et al., 1991). Simultaneous signals in both Greenland and Antarctic ice cores with similar magnitude are consistent with a tropical origin, and analysis of sulfur isotopes in ice cores supports the hypothesis of a major volcanic eruption with stratospheric injection (Cole-Dai et al., 2009).

    Based on ice-core sulfur records from Antarctica and Greenland, the 1809 eruption is estimated to have injected 19.3 $\pm$
3.54 Tg of sulfur (S) into the stratosphere (Toohey and Sigl, 2017). This value is roughly 30% less than the estimate for the 1815 Mt. Tambora eruption, and roughly twice that of the 1991 Pinatubo eruption. Accordingly, the 1809 eruption produced the 2nd largest volcanic stratospheric sulfur injection (VSSI) of the 19th century, and the 6th largest of the past 1000 years. For comparison, the Ice-core Volcanic Index 2 (IVI2) database (Gao et al., 2008) estimates that the 1809 eruption injected 53.7 Tg of sulfate aerosols, which corresponds to 13.4 Tg S. While smaller than the estimate of Toohey and Sigl (2017), the IVI2 value
lies within the reported 2-sigma uncertainty range. Uncertainties in VSSI and related uncertainties in the radiative impacts of the volcanic aerosol could be relevant for the interpretation of post-volcanic climate anomalies, as recently discussed for the 1815 Mt. Tambora eruption and the year without summer in 1816 (Zanchettin et al., 2019; Schurer et al., 2019).

    As well as the location and the magnitude, the timing of the 1809 eruption is also uncertain. Analysis of high-resolution ice-core records points to an eruption in February 1809 $\pm$ 4 months (Cole-Dai, 2010). Observations from South America
of atmospheric phenomena consistent with enhanced stratospheric aerosol (Guevara-Murua et al., 2014) suggest a possible eruption in late November or early December 1808 (4.12.1808 $\pm$ 7 days), although there is no direct link between these observations and the ice-core sulfate signals. Chenoweth (2001) proposed an eruption date of March-June 1808 based on a sudden cooling in Malaysian temperature data and maximum cooling of marine air temperature in 1809. Such uncertainty in the eruption date has implications for the associated spatio-temporal pattern of aerosol dispersal and hemispheric and global
climate impacts (Toohey et al., 2011; Timmreck, 2012). The climatic impacts of the 1809 eruption have been mostly studied in the context of the early 19th century volcanic cluster (e.g., Cole-Dai et al., 2009; Zanchettin et al., 2013, 2019; Anet et al., 2014; Winter et al., 2015; Brönnimann et al., 2019a) or of multi-eruption investigations (e.g., Fischer et al., 2007; Rao et al., 2017). Lesser attention has been placed on characterizing and understanding the short-term climatic anomalies that specifically followed the 1809 eruption. Available observations and reconstructions indicate ambiguous signals in NH land-mean summer
temperatures reconstructed from tree-ring data for this period. For example, Schneider et al. (2017) found that, among the ten largest eruptions of the past 2500 years, the 1809 event was one of two that did not produce a significant "break" in the





temperature time series. While the temperature reconstruction reports cooling in 1809/1810, Schneider et al. (2017) note that reconstructed temperatures did not return to their climatological mean after the initial drop and remained low until the Mt. Tambora eruption in 1815. Hakim et al. (2016) presented multivariate reconstructed fields for the 1809 volcanic eruption from
the last millennium climate reanalysis (LMR) project. They found abrupt global surface cooling in 1809 which was reinforced in 1815. The post volcanic global-mean 2m temperature anomalies show however a wide spread of up to 0.3 °C in the LMR between ensemble members and experiments using different combinations of calibration data for the proxy system models and prior data in the reconstruction. Using the LMR paleoenvironmental data assimilation framework, Zhu et al. (2020) demonstrate that some of the known discrepancies between tree-ring data and paleoclimate models can partly be resolved by assimilating
tree-ring density records only and focussing on growing-season temperatures instead of annual temperature while performing the comparison at the proxy locales. However differences remain for large events like the Mt. Tambora 1815 eruption.

In this study, we investigate the climate impact of the 1809 eruption by using Earth system model ensemble simulations and by analyzing new and existing observational and proxy based data sets. We explore how uncertainties in the magnitude and spatial structure of the forcing propagate to the magnitude and ensemble variability of post-eruption regional and hemispheric
climate anomalies.

In section 2, we briefly describe the applied methods, model, experiments and data sets. Section 3 provides an overview of the reconstructed and observed climate effects of the 1809 eruption, while section 4 presents the main results of the model experiments including a model-data intercomparison. The results are discussed in section 5. The paper ends with a summary and conclusions (section 6).

## 2  Methods and Data

### 2.1  Methods

#### 2.1.1  Model

We use the latest low-resolution version of the Max-Planck-Institute Earth-System-Model (MPI-ESM1.2-LR, Mauritsen et al. (2019)), an updated version of the MPI-ESM used in the Coupled Model Intercomparison Project Phase 5 CMIP5 (Giorgetta
et al., 2013). The applied MPI-ESM1.2 configuration is one of the two reference versions used in the Coupled Model Intercomparison Project Phase 6 (CMIP6, see Eyring et al., 2016). It consists of four components: the atmospheric general circulation model ECHAM6 (Stevens et al., 2013), the ocean-sea ice model MPIOM (Jungclaus et al., 2013), the land component JSBACH (Reick et al., 2013) and the marine model HAMOCC (Ilyina et al., 2013). In MPI-ESM1.2, ECHAM6.3 is used, which includes modifications of the convective mass flux, convective detrainment and turbulent transfer, the fractional cloud cover and a new
representation of radiative transfer with respect to its CMIP5 version. ECHAM6.3 is run with a horizontal resolution in the spectral space of T63 (~200 km) and with 47 vertical levels up to 0.01 hPa with 13 model levels above 100 hPa. The MPIOM, which is run in its GR15 configuration with a nominal resolution of 1.5°around the equator and 40 vertical levels, has remained largely unchanged with respect to the CMIP5 version. Several revisions with respect to the MPI-ESM CMIP5 version have





however been made for the land physics and biogeochemistry components and the ocean carbon cycle model component. A

detailed description of all updates is given in Mauritsen et al. (2019). Previous studies have successfully shown the MPI-ESM is especially well-suited for paleo-applications and has been widely tested and employed in the context of the climate of the last millennium (e.g., Jungclaus et al., 2014; Zanchettin et al., 2015; Moreno-Chamarro et al., 2017).

### 2.1.2   Forcing

The applied volcanic forcing is compiled with the Easy Volcanic Aerosol (EVA) forcing generator (Toohey et al., 2016). EVA

provides an analytic representation of volcanic stratospheric aerosol forcing, prescribing the aerosol's radiative properties and primary modes of their spatial and temporal variability. This permits the compilation of physically consistent forcing estimates also for historic eruptions. EVA uses sulfur dioxide ($SO_2$) injection time series as input and applies a parameterized three-box model of stratospheric transport to reconstruct the space–time structure of sulfate aerosol evolution. Simple scaling relationships serve to construct stratospheric aerosol optical depth (SAOD) at 0.55 $\mu$m and aerosol effective radius from the

stratospheric sulfate aerosol mass, from which wavelength-dependent aerosol extinction, single scattering albedo, and scattering asymmetry factors are derived for pre-defined wavelength bands and latitudes. Volcanic stratospheric sulfur injection (VSSI) values for the simulations performed in this work are taken from the eVolv2k reconstruction based on sulfate records from various ice cores from Greenland and Antarctica (Toohey and Sigl, 2017). Compared to prior volcanic reconstructions, eVolv2k includes improvements of the ice core records in terms of synchronization and dating, as well as in the methods used

to estimate VSSI from them.

Consistent with the estimated range given by Cole-Dai (2010) and the convention for unidentified eruptions used by Crowley and Unterman (2013), the eruption date of the unidentified 1809 eruption is set to occur on the 1st of January 1809 and located at the equator. The eVolv2k best estimate for the VSSI of the 1809 eruption is 19.3 Tg S with a 1$\sigma$ uncertainty of $\pm$ 3.54 Tg S, based on the variability between individual ice-core records and model-based estimates of error due to the limited hemispheric

sampling provided by ice sheets. To incorporate this uncertainty into climate model simulations, we constructed aerosol forcing time series using the central (or best) VSSI estimate, as well as versions which perturbed the central estimate by adding and subtracting two times the estimated uncertainty ($\pm 2\sigma$) from the central VSSI estimate. These three forcing sets are hereafter termed "Best", "High", and "Low", respectively. Constructed in this manner, the range from "Low" to "High" forcing should roughly span a 95% confidence interval of the global-mean aerosol forcing.

There are other important sources of uncertainty in the reconstruction of stratospheric aerosol, further to that related to the magnitude of the sulfur deposition. For example, the transport of aerosol from the tropics to each hemisphere has been seen to be quite variable for the tropical eruptions of Pinatubo in June 1991, El Chichón in April 1982 and Agung in March 1963, which likely arises due to the particular meteorological conditions at the time of the eruption (Robock, 2000). While the 1991 Pinatubo eruption produced an aerosol cloud that spread relatively evenly to each hemisphere, the aerosol from the

1982 El Chichón eruption and the 1963 Agung eruption were heavily biased to one hemisphere (Suppl. Fig. S1). Furthermore, the lifetime, evolution and spatial structure of aerosol properties may vary significantly as a result of the injection height of the volcanic plume (Toohey et al., 2019; Marshall et al., 2019). Recently, Yang et al. (2019) pointed out that an accurate

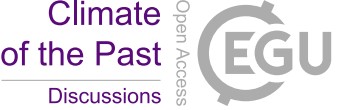



**Figure 1.** Volcanic radiative forcing. Global stratospheric aerosol optical depth (SAOD) at 0.55 $\mu$m based on eVolv2k VSSI estimates (Toohey and Sigl, 2017) and calculation with the volcanic forcing generator EVA (Toohey et al., 2016) for the four different forcing scenarios: "Best", "Low", "High" and "nNHP" for the 1809 eruption and the "Best" forcing scenario for the Mt. Tambora eruption. Bottom: Spatial and temporal distribution of a zonal mean stratospheric SAOD for the four experiments.

reconstruction of the spatial forcing structure of volcanic aerosol is important to get a reliable climate response. In fact, given the current knowledge, an interhemispheric symmetric forcing is only one of several realistic possibilities for the 1809 eruption.

In recognition of these additional sources of uncertainty (and motivated in part by the apparent muted temperature response





to the 1809 eruption in NH tree ring records) we constructed a fourth forcing set, which is identical to the "Best" forcing in the tropics and in the Southern Hemisphere (SH), but has the aerosol mass in the NH extratropics completely removed. Such forcing may seem unrealistic given the significant sulfur deposition to Greenland from the 1809 eruption, but we surmise that weak NH radiative forcing (relative to the Greenland sulfur signal) could result if the aerosol cloud had a short lifetime in the NH, especially if transport and removal of aerosol occurred primarily in the boreal winter when solar insolation is small in the NH. This forcing scenario, which we call "no-NH plume" or "nNHP" in the following, should be interpreted as a rather extreme "end-member" in terms of NH forcing. Time series of global-mean and zonal-mean SAOD at 0.55 $\mu$ for the different 1809 aerosol forcing scenarios discussed above are shown in Fig. 1, together with the "Best" scenario after the Mt. Tambora eruption. Peak global-mean SAOD values following the 1809 eruption range from 0.17 to 0.33 from the "Low" to "High" scenarios, roughly corresponding to forcing from a little stronger than that from the 1991 Pinatubo eruption to a little weaker than the 1815 Mt. Tambora eruption, respectively. The "nNHP" scenario produces a global-mean SAOD that peaks at a value of 0.21, i.e., in between the "Low" and "Best" scenarios, and decays in a manner very similar in magnitude to the "Low" scenario. The latitudinal spread of aerosol is relatively evenly split between the NH and SH in the "Best", "Low", and "High" scenarios, with offsets in the timing of the peak hemispheric SAOD resulting from the parameterized seasonal dependence of stratospheric transport in EVA. After the removal of aerosol mass from the NH extratropics in the construction of the "nNHP" scenario, the SAOD is predictably negligible in the NH extratropics and a strong gradient in SAOD is produced at 30°N.

### 2.1.3 Experiments

We have performed ensemble simulations of the early 19th century with the MPI-ESM1.2-LR for each of the four forcing scenarios for the 1809 eruption (*Best*, *High*, *Low* and *nNHP*). All simulations also include the eVolv2k "Best" forcing estimate for the Mt. Tambora eruption from 1815 onwards. Related experiments using a range of different forcing estimates for the 1815 Mt. Tambora eruption were used in Zanchettin et al. (2019) and Schurer et al. (2019) to investigate the role of volcanic forcing uncertainty in the climate response to the 1815 Mt. Tambora eruption, in particular the "year without summer" in 1816. For each experiment we have produced 10 realizations branched off every 100 to 200 years from an unperturbed 1,200-year-long pre-industrial control run (constant forcing, excluding background volcanic aerosols) to account for internal climate variability. All simulations were initialized on 1st January 1800 with constant preindustrial forcing except for stratospheric aerosol forcing.

## 2.2 Data

### 2.2.1 Temperature reconstructions

**Tropical temperature reconstructions**

In our study, we compare three different sea-surface temperature (SST) reconstructions with the MPI-ESM simulations. The temperature reconstruction TROP is a multi-proxy tropical (30°N–30°S, 34°E-70°W) annual SST reconstruction between 1546–1998 (D'Arrigo et al., 2009). TROP consists of 19 coral, tree-ring and ice-core proxies located between 30°N and 30°S. The records were selected on the basis of data availability, dating certainty, annual or higher resolution, and a documented





relationship with temperature. It shows annual-to multi decadal-scale variability and explains 55% of the annual variance in the most replicated period 1897–1981. Further, 400-year long spatially resolved tropical SST reconstructions for four specific

regions: the Indian Ocean (20°N–15°S, 40–100°E), the western (25°N–25°S, 110–155°E) and eastern Pacific (10°N–10°S, 175°E–85°W), and western Atlantic (15–30°N, 60–90°W) were compiled by Tierney et al. (2015) based on 57 published and publicly archived marine paleoclimate data sets. The four regions were selected based on the availability of nearby coral sampling sites and an analysis of spatial temperature covariance. An even more regionally specific SST reconstruction was developed by D'Arrigo et al. (2006) for the Indo-Pacific warm pool region (15°S–5°N, 110–160°E) using annually resolved

teak ring width and coral $\delta^{18}$O records. This September–November mean SST reconstruction dates from A.D. 1782–1992 and explains 52% of the SST variance in the most replicated period. This record was used in the D'Arrigo et al. (2009) TROP reconstruction.

**Northern Hemisphere extratropical temperature reconstruction**

We compare our climate simulations of the early 19th century with four near-surface air temperature (SAT) reconstruc-

tion, which have all been used to assess the impacts of volcanic eruptions on surface temperature. The N-TREND (Northern Hemisphere Tree-Ring Network Development) reconstructions (Wilson et al., 2016; Anchukaitis et al., 2017) are based on 54 published tree-ring (TR) records and use different parameters as proxies for temperature. Eleven of the records are derived from ring-width (RW), 18 from maximum latewood density (MXD), and 25 are mixed records which consist of a combination of RW, MXD, and blue intensity (BI) data (see, Wilson et al., (2016) for details). The N-TREND database domain covers the NH

midlatitudes between 40°and 75°N with at least 23 records extending back to at least AD 978. Two versions of the N-TREND reconstructions are used herein. N-TREND (N), detailed in Wilson et al. (2016), is a large-scale mean composite May-August temperature reconstruction derived from averaging the 54 TR records weighted to 4 longitudinal quadrats, with separate nested calibration and validation performed as each shorter record is removed back in time. N-TREND (S), detailed in Anchukaitis et al. (2017), is a spatial reconstruction of the same season, derived by using Point-by-Point Multiple Regression (Cook et al.,

1994) of the TR proxy records available within 1000 to 2000 kms of the centre point of each 5x5 degree instrumental grid cell. For each gridded reconstruction a similar nesting procedure was used as Wilson et al. (2016). Herein, we use the average of all the grid point reconstructions for the periods where the validation Reduction of Error (RE - Wilson et al. 2006) was greater than zero. The NVOLC reconstruction (Guillet et al., 2017) is a NH summer temperature reconstruction over land (40°–90°N) composed of 25 TR chronologies (12 MXD, 13 TRW) and 3 isotope series from Greenland ice cores (DYE3, GRIP, Crete).

NVOLC was generated using a nested approach and includes only chronologies which encompass the full time period between today and the 13th century. The temperature reconstruction by Schneider et al. (2015) is based on 15 MXD chronologies distributed across the NH extratropics. All the temperature reconstructions show distinct short-time cooling after the largest eruptions of the Common Era. However, Schneider et al. (2017) point to a notable spread in the post-volcanic temperature response across the different reconstructions. This has various possible explanations, including the different parameters used,

the spatial domain of the reconstruction, the method(s) used for detrending, and choices made in the network compilations.



### 2.2.2 Observed temperatures

**Surface air temperature from English East India Company ship logs**

Brohan et al. (2012) compiled an early observational data set of weather and climate between 1789 and 1834 from records of the English East India Company (EEIC), which are archived in the British Library. The records include 891 ships' logbooks

of voyages from England to India or China and back containing daily instrumental measurements of temperature and pressure, as well as wind-speed estimates. Several thousands of weather observations could be gained from these ship voyages across the Atlantic and Indian Oceans providing a detailed view of the weather and climate in the early 19th century. Brohan et al. (2012) found that mean temperatures expressed a modest decrease in 1809 and 1816 as a likely consequence of the two large tropical volcanic eruptions during the period. Following Brohan et al. (2012), here we calculate temperature anomalies from the SAT

measurements recorded in the EEIC logs. We account for the relatively sparse and irregular spatial and temporal sampling by computing for each measurement its anomaly from the HadNMAT2 night marine air temperature climatology (Kent et al., 2013). The SAT anomalies were then binned according to month/year and location and averaged. We present the data as mean temperature anomalies for the tropics (20°S to 20°N) in monthly or annual means. To quantify the impact of the 1809 eruption, anomalies are referenced to the 1800-1808 time period.

**Station data**

Climate model output is compared with monthly temperature series from land stations that cover the period 1806-1820 from a number sources, as compiled in Brönnimann et al. (2019b). The sources include data available electronically from the German Weather Service (DWD), the Royal Dutch Weather service (KNMI), the International Surface Temperature Initiative (Rennie et al., 2014), and the Global Historical Climatology Network (Lawrimore et al., 2011). In addition, we added nine

series digitized from the compilation of Friedrich Wilhelm Dove that were not contained in any of the other sources (Dove, 1838, 1839, 1842,1845). Of the 73 series obtained, 20 had less than 50% data coverage within the period 1806 and 1820 and were thus not further considered. The remaining 53 time series (see appendix table A1) were deseasonalized based on the 1806-1820 mean seasonal cycle and grouped by region (see appendix table A2).

### 2.3 Analysis of model output

Post-eruption climatic anomalies in the volcanically-forced ensembles are compared with both anomalies from the control run (describing the range of intrinsic climate variability), and with anomalies from a set of proxy-based reconstructions and instrumental observations, providing a reference/target to evaluate the simulation under both volcanically-forced and unperturbed conditions. Comparison between the volcanically-forced ensembles and the control run is based on the generation of signals in the control simulation analogous to the post-eruption ensemble-mean and ensemble-spread anomalies. In practice, a large

number (1000) of surrogate ensembles is sampled from the control run, each identified by a randomly chosen year as reference for the eruption. Ensemble means and spreads (defined by 5th and 95th percentiles) of such surrogate ensembles provide an empirical probability distribution that is used to determine the range of intrinsic variability, which is illustrated with the associated 5th-95th percentile ranges. Differences between the volcanically-forced ensembles and the surrogate ensembles are tested





statistically through the Mann-Whitney U test (following, e.g., Zanchettin et al., 2019). When the ensembles are compared
with a one-value target, either an anomaly from reconstructions/observations or a given reference (e.g., zero), significance of
the difference between the ensemble and the target is determined based on whether the latter exceeds a given percentile range
from the ensemble (e.g., the interquartile or the 5th-95th percentile range) or, alternatively, based on a t-test.

Integrated spatial analysis between the simulations and the N-TREND (S) gridded reconstruction is performed through a
combination of the Root Mean Square Error (RMSE) and spatial correlation. Both metrics are calculated by including grid
points in the reconstructions that correspond to the proxy locations and interpolating the model output to those locations with a
nearest-neighbour algorithm. The relative contribution of each location is weighted by the cosine of its latitude to account for
differences in the associated grid-cell area.

We consider four climate indices, namely the Pacific/North American pattern (PNA) index, the North Atlantic Oscillation
(NAO) index, and the North Polar Index (NPI) to describe NH (winter) variability and the Southern Oscillation Index (SOI),
to describe tropospheric variability. Definitions and indices are the same as described in Zanchettin et al. (2015), see appendix
for further details. For each simulation, indices are standardized based on the average and standard deviation calculated for the
period 1800-1808. If not mentioned otherwise, anomalies in the volcanically-forced ensembles are defined as deviations with
respect to the corresponding 1800-1808 average.

## 3   The 1809 eruption in climatic observations and proxy records

In proxy and instrumental records of tropical temperatures, cooling in the years 1809-1811 is generally on par with that after the
1815 Mt. Tambora eruption. Based on annually-resolved temperature-related records from corals, TRs and ice cores, D'Arrigo
et al. (2009) report peak tropical cooling of -0.77 °C in 1811, compared to -0.84 °C in 1817 (Fig. 2a). Tropical SST variability is
modulated by El-Nino Southern Oscillation (ENSO) variability such as neutral to La Nina-like conditions in 1810 and El Nino-
like ones in 1816 (Li et al., 2013; McGregor et al., 2010). The lagged response to the 1809 and 1815 eruptions in the TROP
245 reconstruction is therefore most likely a result of an overlaying El Nino signal. Removing the ENSO signal from the TROP
reconstructions led to a shift of maximum post volcanic cooling from 2 years after the eruption to one year (D'Arrigo et al.,
2009). A clear signal is found in reconstructed Indo-Pacific warm pool SST anomalies from the post-1809 period 1809–1812,
with values of -0.28 °C, -0.73 °C, - 0.76,°C and -0.79°C; compared to -0.30 °C, -0.51°C and -0.51 °C for the post-Tambora
period 1815 – 1817 (D'Arrigo et al., 2006). Chenoweth (2001) reports pronounced tropical cooling from ship-based marine
250 SAT measurements in 1809 (-0.84 °C), similar to that in 1816 (-0.81 °C). More recent analysis of a larger set of ship-based
marine SAT records from the EEIC by Brohan et al. (2012) suggests a more modest cooling for the two early 19th century
eruptions, of about 0.5 °C (Fig. 2b). However, the cooling is again found to be of comparable magnitude after the 1809 and
1815 eruptions.

TR records capture volcanically-forced summer cooling very well (e.g., Briffa et al., 1998; Hegerl et al., 2003; Schneider
et al., 2015; Stoffel et al., 2015). However, in the NH extratropics, SAT anomalies after 1809 are more spatially and temporally
complex compared to the typical post-eruption pattern with broad NH cooling. In TR-based temperature reconstructions for

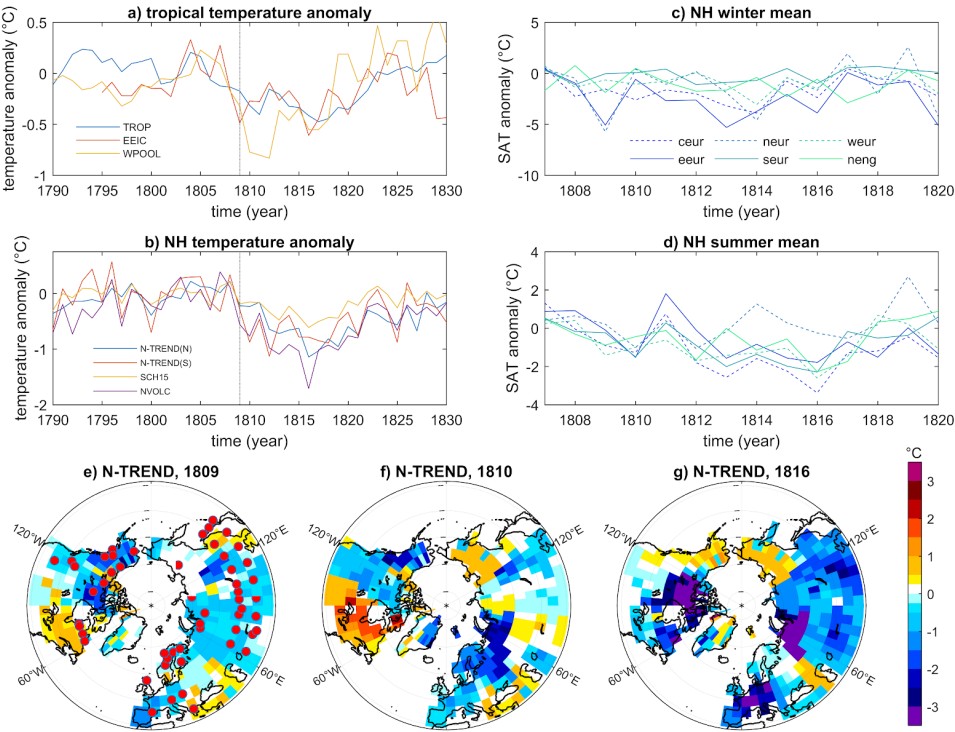

**Figure 2.** Observed and reconstructed temperature anomalies around the 1809 volcanic eruption. a) Reconstructed tropical (30°N–30°S, 34°E-70°W) sea surface temperature (TROP, D'Arrigo et al., 2009), measured tropical marine surface air temperatures from EEIC ship logs (Brohan et al., 2012) and Indo-Pacific warm pool data (D'Arrigo et al., 2006). b) NH summer land temperatures from four tree-ring based reconstructions (Wilson et al. 2016 (N-TREND(N)), Anchukaitis et al., 2017 (N-TREND (S)), Guillet et al., 2017 (NVOLC), Schneider et al., 2015 (SCH15)). c-d) Monthly mean NH winter (c) and summer (d) temperature anomalies (°C) from 53 station data averaged over different European regions (Central Europe (CEUR: 46.1-52.5°N, 6-17.8°E), Eastern Europe (EEUR: 47-57°N, 18-32°E), Northern Europe (NEUR: 55-66°N, 10-31°E), Southern Europe (38-46°N, 7-13.5°E), Western Europe (WEUR: 48.5-56°N, 6°W-6°O) and New England (NENG: 41-44°N, 73-69°W). e-g) Mean surface temperature anomalies (°C) for boreal summers of 1809 (e), 1810 (f) and 1816 (g) in NH TR data N-TREND (S) (Anchukaitis et al., 2017). Red dots in panel e) illustrate the location of the tree-ring proxies used in the N-TREND reconstructions.

interior Alaska/Yukon (Briffa et al., 1994; Davi et al., 2003; Wilson et al., 2019), 1810 is one of the coldest summers identified over recent centuries. In earlier reconstructions of summer SAT in different regions of the western United States (Schweingruber et al., 1991; Briffa et al., 1992), 1810 was shown to be the third coldest summer in the British Columbia/Pacific Northwest

region. Likewise, European tree-ring records show cooling after 1809 (e.g., Briffa et al., 1992; Wilson et al., 2016). In contrast, tree-ring networks in certain regions such as Eastern Canada show minimal response after 1809 (Gennaretti et al., 2018).

Based on compilations of regional records, TR-based reconstructions of NH-mean land summer SAT show a large spread in hemispheric cooling after the 1809 eruption (Fig. 2b), with anomalies of -0.87 °C, -0.77 °C, -0.21°C and -0.15 °C in 1810





for the N-TREND (S), NVOLC, N-TREND (N) and SCH15 reconstructions, respectively). Although using the same data set,
the spatial N-TREND (S) and the nested N-TREND (N) reconstructions show a quite different behaviour. In N-TREND (S),
the nature of the spatial multiple regression modelling biases the input records to those that correlate most strongly with local
temperatures which, when available, were are likely MXD data. In all four reconstructions, NH temperature does not return to
the climatological mean after an initial drop in 1810, but remains low, or even exhibits a continued cooling trend until the Mt.
Tambora eruption in 1815 (Schneider et al., 2015). The spatial variability of the reconstructed NH extratropical temperature
response to the 1809 eruption is illustrated in Fig. 2d,e, based on the spatially resolved N-TREND (S) reconstruction (An-
chukaitis et al., 2017), displaying zonal oscillations consistent with a "wave-2" structure, that are especially evident in 1810
but already appreciable in 1809. This hemispheric structure is in contrast with the relatively uniform cooling seen in TR records
for Tambora (Fig. 2f) and indeed for many of the largest eruptions of the past millennium (Hartl-Meier et al., 2017).

Information about regional and seasonal mean NH temperature anomalies in the early 19th century can be obtained from
different station data across Europe and from New England (Fig. 2c). In NH winter the measurements reflect the high variability
of local scale weather (Fig. 2c). Warm anomalies, an indication for post-eruption "NH winter warming", are clearly visible in
1816/1817 in the 2nd winter after the Mt. Tambora eruption in 1815. Northern Europe shows the largest warm anomaly for all
regions (about 3 °C). Warm NH winter anomalies between 1.5°C and 2 °C are seen in the winter 1809/1810 over Northern
and Eastern Europe and over New England. Strong cooling is however found for the 1808/1809 winter in Northern and Central
Europe. NH summer temperature anomalies are less variable than in winter (Fig. 2d). A local distinct cooling is found in
the "year without summer" in 1816 over all regions except Northern Europe, where it occurs a year later. The cooling after
the 1809 eruption is not so pronounced as after the Mt. Tambora eruption in 1815. Station data over Northern, Eastern and
Southern Europe show a local minimum in 1810, which does not appear over Western and Central Europe and New England.
Interestingly, warm anomalies in the order of 2 °C are found in summer 1811 over Central and Eastern Europe.

## 4   Results

### 4.1   Simulations

Firstly, we compare the simulated evolutions of monthly mean near-surface (2m) air temperature anomalies between the four
experiments globally, in the tropics and NH extratropics (Fig. 3). Ensemble mean global mean temperature anomalies grow
through 1809, and reach peak values through 1810 in all experiments, before decaying towards climatological values (Fig. 3a).
Peak cooling reaches around 1.0 °C in the *High* experiment, compared to  0.5 °C in the *Low* and *nNHP* experiments. Peak
temperature anomalies across the experiments correlates with the magnitude of prescribed AOD (Fig. 1a), and the responses
are qualitatively consistent with expectations, with the AOD for the *Low* and *nNHP* experiments, which is similar in magnitude
to that from the observed 1991 Pinatubo eruption, leading to global mean temperature anomalies also similar to those observed
after Pinatubo. Global ensemble mean near-surface temperature anomalies are close together in *Low* and *nNHP* over boreal
summer but differ for boreal winter when the intrinsic variability is higher. *Low* is the only experiment for which large-scale
temperatures return within the 5th-95th percentile range percentile interval of the control run before the Mt. Tambora eruption



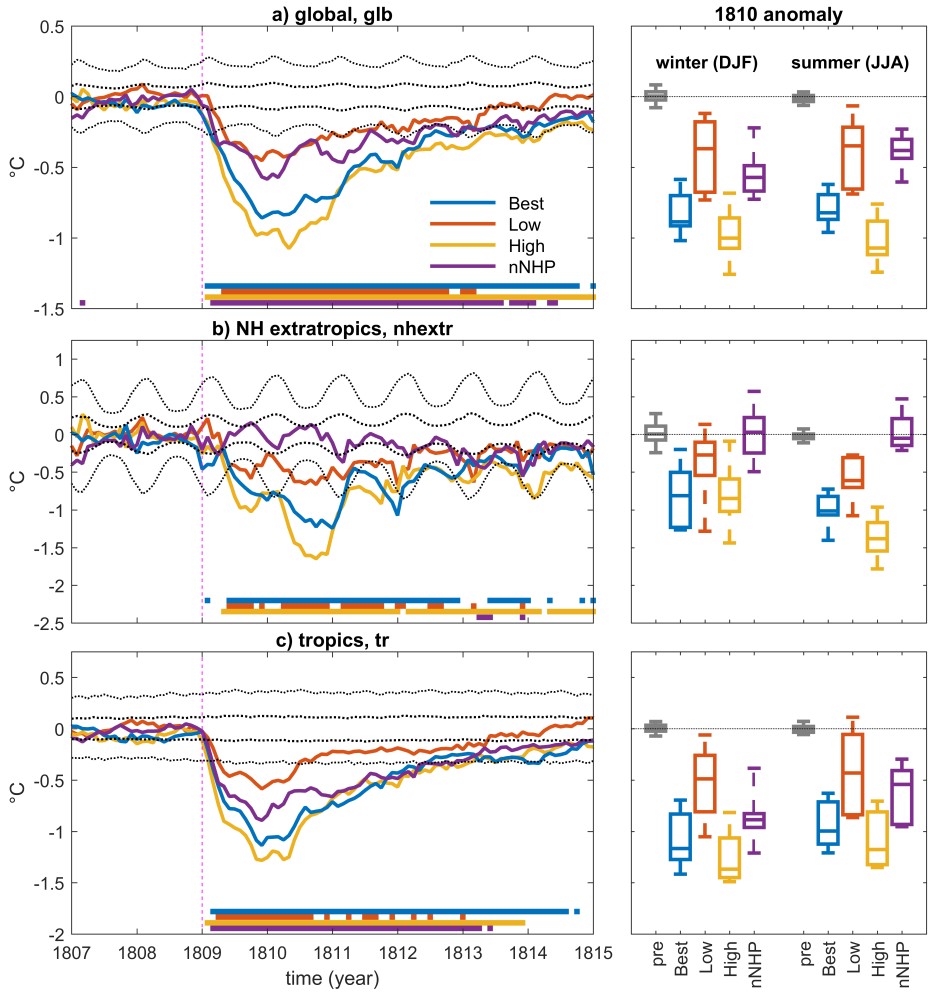

**Figure 3.** Global, tropical and extratropical temperature anomalies. Left: Simulated ensemble-mean monthly anomalies of (a) global, b) extra-tropical northern hemisphere and c) tropical averages of near-surface air temperature with respect to the pre-eruption (1800-1808) climatology. All data are deseasoned using the respective annual average cycle from the control run. Thick (thin) black dashed lines are the 5th–95th percentile intervals for signal occurrence in the control run for the ensemble mean (ensemble spread). Bottom bars indicate periods when an ensemble member's monthly mean temperature (color code as for the time series plots) is significantly different (p = 0.05) from the control run according to the Mann-Whitney U test. Right: ensemble distributions (median, 25th-75th and 5th-95th percentile ranges) of seasonal mean anomalies for the first post-eruption winter (1809-1810, DJF) and summer (1810, JJA) following the 1809 eruption as well as for the pre-eruption period (1800-1808).

in 1815. Global mean temperature anomalies of the other three experiments return only within the 5th-95th percentile range of unperturbed variability by 1815. As expected, almost no significant near-surface temperature anomalies are found for the *nNHP* simulation in the NH extratropics except a few months in spring and autumn 1813 (Fig. 3b). The *nNHP* ensemble-mean





values stay within the interquartile range of the control run but show a slight negative trend between 1809 and 1815. *nNHP* is also the only experiment where the NH extratropical summer of 1814 is colder than the summer of 1809. Internal variability is relatively high in the NH extratropics in particular in NH winter spanning more than 1.5 °C. So, even the ensemble mean near surface temperature anomalies for *Best* and *High* almost reach the 5th-95th percentile range of the control run in the 1st post volcanic winters. Peak cooling appears for all experiments except *nNHP* in the summer 1810. In the tropics, *Best*, *High* and

*nNHP* are outside the 5th-95th percentile range in the first four -post volcanic years while *Low* exceeds the 5th-95th percentile range only for two years (Fig. 3c).

     The ensemble distributions for the seasonal mean of winter 1809/1810 and summer 1810 illustrate the differences between the four experiments not only in the mean anomaly but also for the ensemble spread (Fig. 3 d-f). While for example in summer 1810 the global and tropical ensemble mean of *Low* and *nNHP* are quite close, the ensemble spread is much larger in *Low*

compared to *nNHP*. *Low* has in general the largest ensemble spread independent of season and hemispheric scale. The clearest separation between the experiments appears in the NH extratropics in summer 1810 (Fig. 3f) in line with Zanchettin et al. (2019), who could show with a k mean cluster analysis on a large ensemble that forcing uncertainties can overwhelm initial-condition spread in boreal summer.

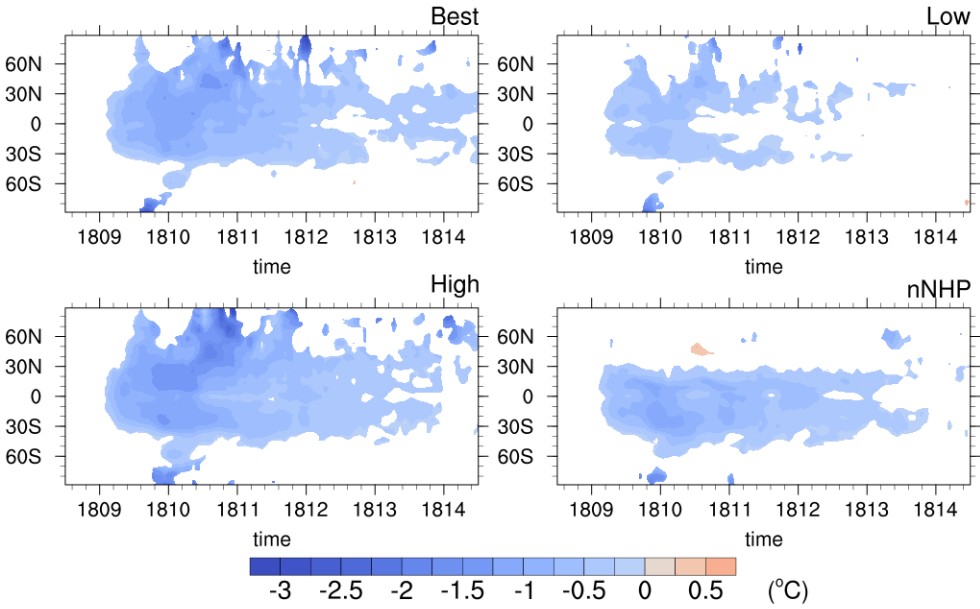

**Figure 4.** Simulated ensemble-mean zonal-mean near-surface air temperature anomalies (°C) for the four MPI-ESM experiments. Only anomalies exceeding one standard deviation of the control run are shown. Anomalies are calculated with respect to the pre-eruption (1800-1808) climatology.

     A more detailed spatial distribution of the simulated temporal evolution of post-volcanic surface temperature anomalies is

seen in the Hovmöller diagram in Fig. 4. It shows that in all four experiments a multiannual surface temperature response is



found in the tropics (30°S-30°N). In the inner tropics, the cooling disappears after one and a half years in *Low* and two to three years later in *Best*, *High* and *nNHP*. In the subtropics, a significant surface cooling signal is found over the ocean until 1815 in *Best* and *High*, while over land no significant cooling appears in 1814 (Fig. S2). A strong cooling signal is found in the NH extra tropics in *Best*, *Low* and *High* in summer 1810 and in *High* and to a small extent also in *Best* in summer 1811. In *nNHP*

no surface cooling is detectable over the NH extratropics in the 1st four years after the eruption, consistent with the prescribed volcanic forcing (see Fig. 1). However, a cooling anomaly is apparent around 60°N in summer 1813, which is seen in the zonal mean over the ocean (Fig. S2) and likely due to decreased poleward ocean heat transport. Significant cooling south of 30°S appears only in austral spring 1809.

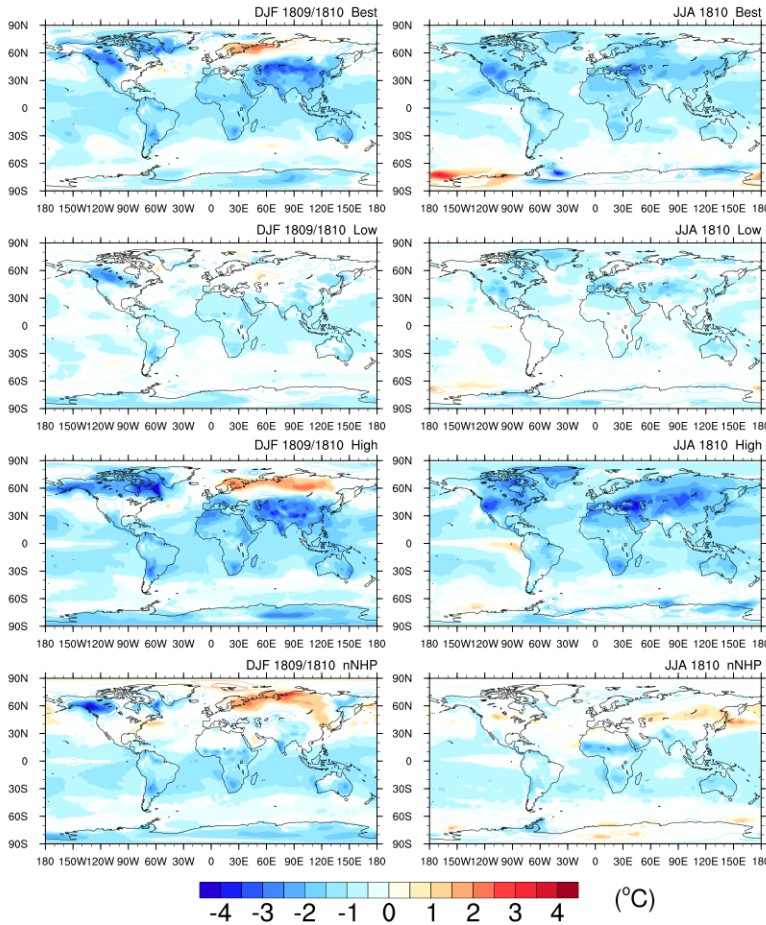

**Figure 5.** Simulated ensemble-mean near-surface air temperature anomalies for the 1st winter (1809/1810) and the second summer (1810) after the 1809 eruption for the four different MPI-ESM simulations. Shaded regions are significant at the 95% confidence level according to a t-test. Anomalies are calculated with respect to the period 1800-1808.



Fig. 5 shows the spatial near-surface air temperature anomalies for the first boreal winter (1809/1810) and the second
boreal summer (1810) after the 1809 eruption for the four experiments. In general, the cooling is strongest over the NH
continents in all experiments revealing a strong cooling pattern over Alaska,Yukon and the Northwest Territories in the first
post-eruption winter. In the *Best* and *High* experiments, relatively strong cold anomalies are found over the Central Asian dry
highland regions around 40°N from the Hindukush in the west to the Pacific, while *Low* and *nNHP* show a small yet significant
cooling over India and Southeast China. In boreal winter a significant warming is visible over Eurasia in all experiments
except for *Low*, where warming anomalies are instead found over the polar ocean, and it is most pronounced in *High* but also
quite extensive in *nNHP*. Such NH winter warming pattern is known to be induced by atmospheric circulation changes (e.g.,
Wunderlich and Mitchell, 2017; DallaSanta et al., 2019) and can occur in post-eruption winters as a dynamic response to the
enhanced stratospheric aerosol layer, when it displays highly variable amplitude of local anomalies (Shindell et al., 2004).
Accordingly, in our simulations the Eurasian winter warming pattern consists of one or two areas with positive temperature
anomalies centered over various locations between Fennoscandia and the Central Siberian Plateau in the different simulations.
Significant cooling, albeit of different strength, is found in the NH extratropics in boreal summer in the three symmetric-forcing
experiments (*Best*, *High*, *Low*). However, while all of them show significant negative temperature anomalies over the North
America continent with a local maximum over California and also cooling over Greenland, no significant anomalies are seen in
*Low* over Fennoscandia. Except for some small regions (Finland, the Kola Peninsula and West Alaska), no significant cooling
is found in *nNHP* in the NH extra tropics in boreal summer. The spatial distribution of the forcing can impact the latitudinal
position of peak surface cooling, which in turn can lead to a shift of the Intertropical Convergence Zone (e.g., Haywood et al.,
2013; Pausata et al., 2020). This is clearly visible in the cold anomaly belt over the Sahel region in the asymmetric forcing
experiment *nNHP*. Significant warm anomalies are detectable in a small band that extends from the Caspian sea to the west to
Japan to the East. Cooling over the ocean is weaker and mostly confined in the tropical belt between 30°S and 30°N. *High* is
the only experiment where a significant El Nino type anomaly is seen over the Pacific Ocean in boreal summer 1810, while in
the other three experiments a slight but not significant warming offshore of the South-American coast appears.

Post-eruption anomalies of select large-scale atmospheric circulation indices from the simulations are shown in Fig.6. We
consider changes in a mode of large-scale atmospheric circulation to be significant when the ensemble distribution of the
associated index excludes the zero value within the interquartile range. Accordingly, for the NAO there is no significant change
from the pre-eruption state for both winter 1809/1810 and summer 1810, with the exception of the *High* ensemble in summer
which has a tendency toward a more negative state of the NAO (Fig. 6a). The NAO reconstruction by Ortega et al. (2015)
indicates a weakly positive anomaly in winter 1809/1810 (not shown). A similar lack of significant post-eruption anomalies is
detected for the PNA, for which only negative winter anomalies from the *nNHP* ensemble and negative summer anomalies for
the *High* ensemble occur (Fig. 6b). Uncertainties affecting currently available PNA reconstructions (Zanchettin et al., 2015;
Franke et al., 2017) prevent deeper insights on the potential role of the PNA for the post-1809 climate anomalies. We therefore
only highlight the overall low signal-to-noise ratio of PNA anomalies in 1810 in all our ensembles. The NPI is strongly inversely
correlated with the PNA, hence there is a significant tendency towards positive post-eruption winter anomalies in *nNHP* (Fig.
6c). Further, significant anomalies of opposite signs are detected in summer, negative for Best and positive for *nNHP*, the



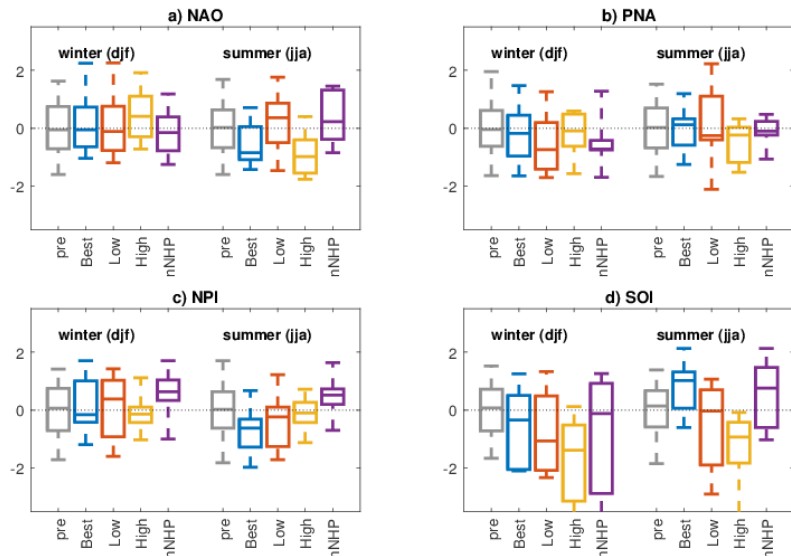

**Figure 6.** Atmospheric circulation indices. Box-Whisker plots (median, 25th-75th and 5th-95th percentile ranges) of seasonal anomalies of circulation indices: a) North Atlantic Oscillation (NAO), b) Pacific/North American pattern (PNA), c) the North Polar Index (NPI) and d) the Southern Oscillation Index (SOI) from the model experiments for the first post-eruption winter (1809/1810, DJF) and second post-eruption summer (1810, JJA) following the 1809 eruption. Pre-eruption (1800-1808) data, shown as grey Box-Whisker plots, are used to standardize the indices (see Methods)

latter with a narrow distribution as shown by the tight whisker plot. Although the PNA is mostly active in NH winter and is

therefore negligible for the NH summer circulation, the weakening of the summer Aleutian Low is expressed by the positive NPI index in *nNHP* which is compatible with the Western North American cooling in the reconstructed temperature pattern (Fig. 3). The SOI shows significant post-eruption anomalies in the *High* ensemble in winter (negative) and in the *High* and *Best* ensembles in summer, the latter with opposite signs (Fig. 6d). Overall, the results illustrate the substantial differences in the post-eruption evolution of continental and subcontinental climates that can be produced by internal climate variability

and forcing structure through changes in the large-scale atmospheric circulation, as seen by the spread of responses within individual ensembles (often as large as the range of pre-eruption variability) and by the possibility of non-overlapping response distributions generated by different forcings (seen for, e.g., the NPI and SOI indices in summer 1810).

## 4.2 Model-data comparison

### 4.2.1 Tropics

A multiannual cooling signal is found in the MPI-ESM simulations in the tropical region after the unidentified 1809 eruption (Figs. 3 and 4). The same signature is detected in the English East India Company (EEIC) ship-based surface air tempera-



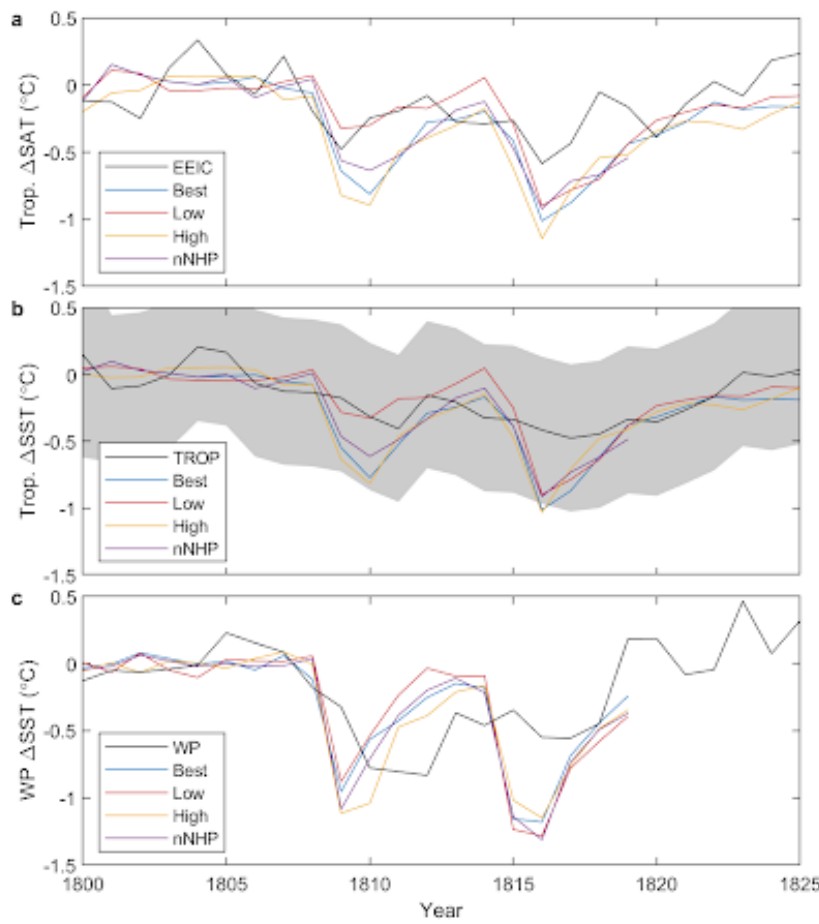

**Figure 7.** Comparison of tropical temperatures anomalies. Comparison of the MPI-ESM simulations with a) tropical and annual mean (30°N-30°S) surface air temperature from shipbourne measurements of the English East India Company (EEIC, Brohan et al., 2012), b) annual mean tropical sea surface temperature (SST) reconstruction (TROP, D'Arrigo et al., 2009) over the tropical IndoPacific (30°N-30°S, 34°E-70°W), c) seasonal mean (Sep-Nov) SST reconstruction (D'Arrigo et al., 2006) anomalies over the Indonesian warmpool (WP, 15°S–5°N, 110–160°E). The black line represents in all panels the observed or reconstructed data while the colored lines represent ensemble means of the respective model simulations. The grey shaded regions in b) indicate the 95% confidence interval of the reconstruction. Anomalies are taken with respect to the years 1800 to 1808.

ture anomaly annual means (Brohan et al., 2012) and in tropical SSTs reconstruction (TROP, D'Arrigo et al., 2009) and the Indo-Pacific warm pool (D'Arrigo et al., 2006) in Fig. 7. The simulated ensemble mean temperatures (Fig. 7a) bracket the observed anomaly in the EEIC data in 1809, with the observed value falling between the results of the *Low* and *nNHP* forcing



experiments.In 1810-1812, the cooling in the *Best*, *High* and *nNHP* experiments is stronger than that observed, and therefore the results from the *Low* experiment are generally most consistent with the ship-borne measurements (Fig. 7a). A comparison of TROP with our four experiments reveals that all experiments lie within the 5th-95th percentile interval of the TROP reconstruction although the reconstructed SST response appears to be dampened in comparison to the model experiments (Fig. 7b). Although the long term trends of TROP and the model experiments are in general agreement, the dampened post volcanic

cooling could reflect autocorrelative biases in the proxies (Lücke et al., 2019). Detailed scrutiny of high resolution tropical SST proxies and their potential biases to robustly reflect volcanically forced cooling has not been made in the same way that has been performed for tree-ring archives over the last decade (Anchukaitis et al., 2012; D'Arrigo et al., 2013; Esper et al., 2015; Franke et al., 2013; Lücke et al., 2019). A similar behavior is found for the Indonesian warm pool (Fig. 7c). However, in contrast to the whole tropics, the differences between the different forcing experiments are much smaller for the warm pool

region compared to the wider tropical regions and the volcanic signal is more pronounced in the reconstructed SST at least for the unidentified 1809 eruption.

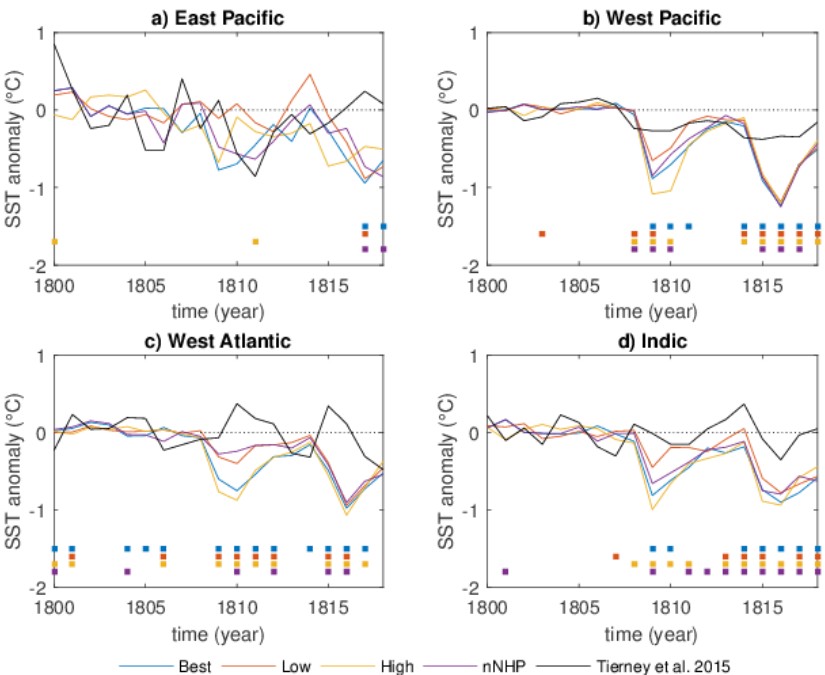

**Figure 8.** Coral-SST comparison. Comparison of reconstructed tropical annual mean SST (Tierney et al., 2015) with the MPI-ESM experiments over: a) eastern Pacific (10°N–10°S, 175°E–85°W), b) western Pacific (25°N–25°S, 110–155°E), c) western Atlantic (15–30°N, 60–90°W) and d) the Indian (20 °N–15°S, 40–100°E) oceans. Black solid line: SST reconstruction; colored lines: ensemble means of the model simulations. Anomalies are taken with respect to the years 1800 to 1808. The squares on the bottom of each panel indicate years where the observation lies outside the simulated ensemble range (color code as for the ensemble mean).





Tierney et al. (2015) provided coral-based reconstructions of tropical SSTs for four different ocean regions: the Indian Ocean, the West and the East Pacific, and the West Atlantic. Comparison of our four experiments with the coral-based reconstructions reveals quite different behaviour and simulation-reconstruction agreement across the various regions (Fig. 8). For the East

Pacific region, the reconstruction and the MPI-ESM simulations are not inconsistent with each other over the 1809 period showing a substantial high variability (Fig. 8a) reflecting the influence of both ENSO and volcanic cooling. A clear volcanic signal is therefore found for the four experiments only for the Mt. Tambora eruption, while for the 1809 eruption, *High* and *Best* show a distinct cooling in 1809 and *nNHP* in 1811. In contrast to the East Pacific, variability in the West Pacific is rather small (Fig. 8b). In all four experiments a clear volcanic signal is visible in the simulated ensemble mean SST anomaly after

the 1809 eruption and the Mt. Tambora eruption, whereas only a weak signal appears for both eruptions in the reconstruction. Interestingly, in the West Atlantic, two distinct positive SST anomalies appear in the reconstructions in the aftermath of the unidentified 1809 and the Mt. Tambora eruption, while the MPI-ESM simulations shows cooling (Fig. 8c). Reasons for the anticorrelated behaviour are not obvious per se and may be related to changes in either ocean circulation or other climate factors than SST that influence the coral record, such as salinity and precipitation. In the reconstruction, the Indian Ocean is the only

region that displays a peak cold anomaly after the Mt. Tambora eruption, but the magnitude of this cooling is comparable to an apparent cooling in 1807. A clear reference to the Mt. Tambora eruption is therefore difficult to establish. No large cooling is found in the coral data after 1809 over the Indian Ocean (Fig. 8d).

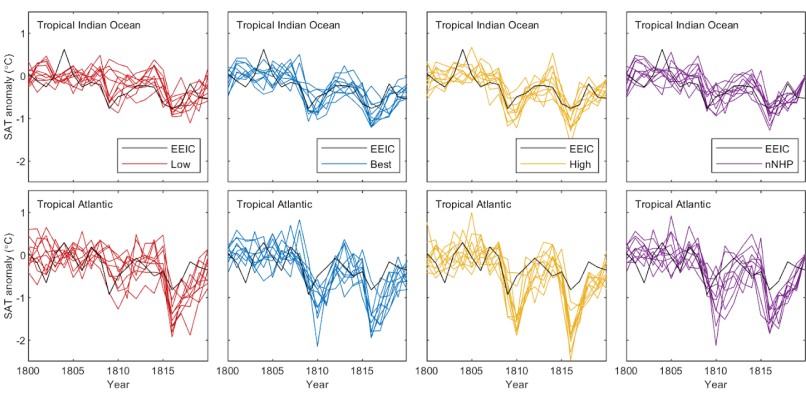

**Figure 9.** Annual mean surface-air temperature anomalies from shipbourne measurements of the English East India Company (EEIC) (Brohan et al., 2012) over the tropical Indian and Atlantic oceans (black line) compared to similarly sampled model simulations from the *Low*, *Best*, *High* and *nNHP* forcing ensembles as labeled. Anomalies are taken with respect to the years 1800 to 1808.

Instrumental measurements from the tropical region are sparse and no continuous temperature record covering the early 19th century exists. Fig. 9 shows a comparison between the model simulations with ship-based surface air temperature measure-

ments from the tropical Atlantic and Indian Oceans. For each ocean basin, the model output is sampled at the locations and times of the ship measurements. For the Indian Ocean, observed temperature anomalies after 1809 are within the model ensem-





ble spread of all the model ensembles. The model response in the Indian Ocean is quite variable for the *Low* forcing experiment, with some members showing no apparent cooling, and others with cooling of up to 0.9 °C. Overall, observed Indian Ocean temperature anomalies are on the lower edge of the *Low* ensemble. For the *Best*, *High* and *nNHP* experiments, the simulated

cooling over the Indian Ocean is more consistent across individual simulations, with the ensemble spread enveloping the observed temperature time series. While *Low* forcing is not inconsistent with the observed Indian ocean temperatures, *Best*, *High* or *nNHP* appear more likely scenarios. In the tropical Atlantic, the observed cooling after the 1809 eruption is slightly stronger than for Mt. Tambora, and slightly stronger than that in the Indian Ocean. The maximum observed cooling in 1809 is roughly within the spread of all the model ensembles. However, while observed tropical Atlantic temperature anomalies are largest in

1809, the simulated cooling usually peaks in 1810. In 1810, the observed cooling is less than simulated in the *Best* ensemble, and smaller than all but one of the individual simulations in the *High* ensemble. Looking at the years after the Mt. Tambora eruption, simulations and observations agree relatively well in the Indian Ocean, while in the Atlantic, the model simulations overestimate the post-Tambora cooling. Since satellite observations of the aerosol cloud from the 1991 eruption of Pinatubo show that aerosol quickly spreads uniformly across the tropics, it is unlikely that aerosol forcing from the 1809 eruption would

be significantly different between the tropical Indian and Atlantic ocean basins. Therefore, differences in temperature response in the model between the two regions seems more likely to be related to model sensitivity, which might particularly be linked to differences in ocean circulation and/or mixed layer depth.

### 4.2.2 Northern hemisphere extratropics

Fig. 10 shows a comparison between the model experiments with four NH summer land near surface temperature reconstruc-

tions from TR records, including the nested N-TREND (N) (Wilson et al., 2016) and the spatial N-TREND (S) reconstruction (Anchukaitis et al., 2017). To ensure comparability between the reconstructions and the model results, the data are expressed as anomalies w.r.t. 1800-1808. The *High* and *Best* experiments show significantly larger cooling than the reconstructions, and are outside the 95% confidence interval of the N-TREND (N) reconstruction. Simulated SAT anomalies in *nNHP* are generally smaller than the reconstructions between 1809 and 1815. Best agreement between the ESM simulations and the data after the

1809 eruption is found for *Low*. In NH summer 1810 and 1811, *Low* matches the reconstructed temperature anomalies from the NVOLC (Guillet et al., 2017) and N-TREND (S) records quite well. Interestingly, the devil really is in the detail. Despite the data richness of this period, the temporal evolution (trend) differs substantially between the different TR reconstructions. In N-TREND (N) the evolution is a step-like temperature decrease with a 1st step in 1809, followed by a second one in 1812 and persistent low values until 1816. Distinct peak cooling appears in NVOLC and N-TREND (S) in NH 1810 followed by a short

recovery phase in 1811 and a drop in 1812, but while summer SAT anomalies stay constant in the NVOLC reconstruction, for N-TREND (S), they start to recover again after 1812. Schneider et al. (2015) show only a small cooling trend between 1809 and 1815. In their reconstruction, temperatures after the 1809 event did not return to their climatological mean after the initial drop, but remained low until the Mt. Tambora eruption in 1815. Compared to the reconstructions, the ESM simulations (*High*, *Best*, *Low*) show a very different temporal evolution with a relatively fast recovery after the 1809 eruption to near background

conditions, followed by a second cooling peak for the Mt. Tambora eruption starting in 1816. In the MPI-ESM model sim-





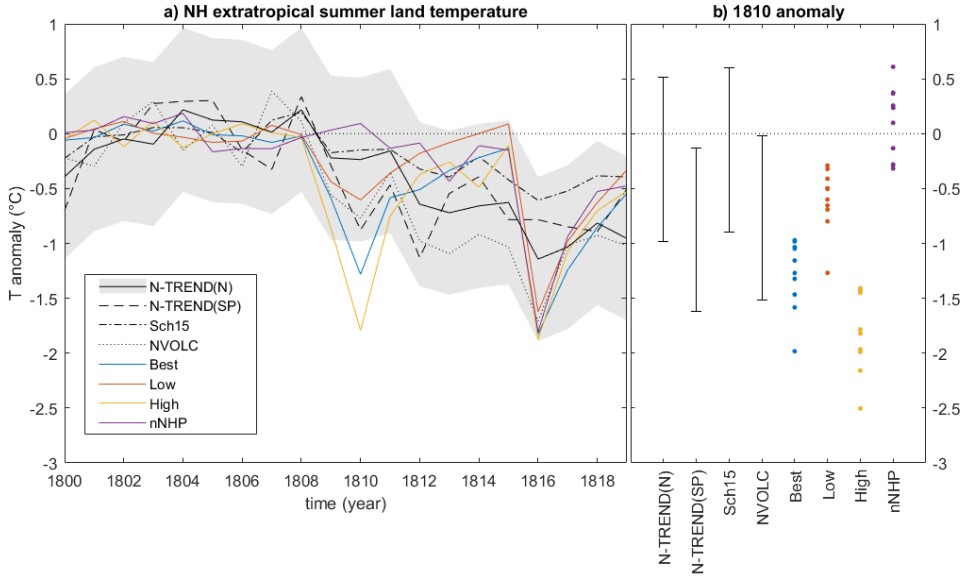

**Figure 10.** Comparison of NH extratropical summer land temperatures. a) Comparison of simulated NH extratropical (40-75°N) summer land temperature anomalies (seasonal and spatial averaged) with four different NH TR-based temperature reconstructions (Wilson et al., 2016 (N-TREND (N)), Anchukaitis et al., 2017 ((N-TREND (S)), Guillet et al., 2017 (NVOLC), Schneider et al., 2015 (SCH15)). Anomalies are taken with respect to the years 1800-1808. The black lines represent the TR records, the colored ones the ensemble mean of the four MPI-ESM model experiments. The shaded grey area indicates the 2 sigma uncertainty range for N-TREND (N). b) Comparison for the reconstructed and simulated anomalies for the year 1810. Uncertainty ranges for all reconstructions are based on the 2-sigma of the N-TREND (N) reconstruction. Simulated anomalies are shown as individual realizations.

ulations no cooling peak appears in the ensemble mean for the summer of 1812 in contrast to the TR records. *nNHP* is the only experiment which shows only a slight cooling trend between 1810 and 1815, appearing closer to Schneider et al. (2015). Between 1813 and 1815, *nNHP* reveals similar temperature anomalies to *Best* and *High*, while *Low* shows smaller anomalies than all other experiments and even attains positive values.

In Fig. 11,we analyse the spatial patterns of the percentiles of the model ensemble into which the reconstruction falls. If the reconstruction lies in the upper range of the distribution of ensemble members, the reconstructed temperature anomalies are warmer than most simulations, i.e. the majority of simulations are colder than the reconstruction. The *High* ensemble (Fig. 11a) is in many locations colder than the reconstructions, but the reconstruction from Central to Northern Europe lies mostly within the interquartile, i.e. the 25th-75th percentile range of the simulations. This behaviour results from the comparison of

the variable local cooling in the individual simulations with highly heterogeneous temperature anomalies in the reconstruction (Fig. 2). Only in a few regions (Central Europe, West Russia, Alaska) are the simulated temperature anomalies much warmer and the reconstructed one below the 25th percentile. The *Best* experiment (Fig. 11b) indicates a similar behaviour as *High* albeit with more regions where the reconstruction lies within the interquartile range of the simulations, e.g., along the west coast of





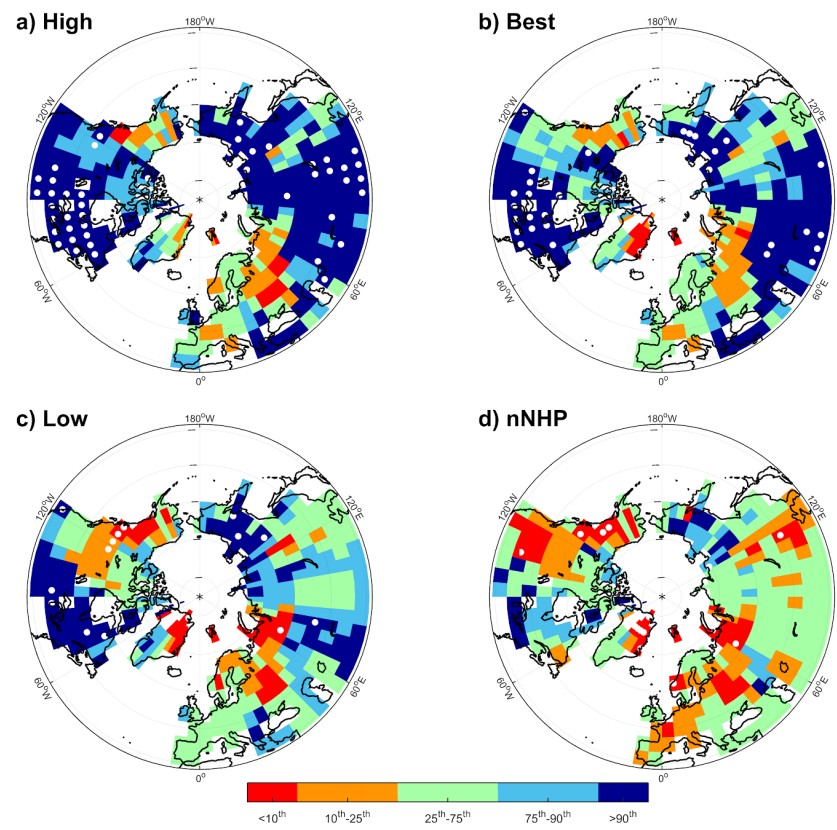

**Figure 11.** Spatial comparison of NH extratropical land temperatures for summer 1810. Statistical comparison of reconstructed surface summer temperatures from N-TREND (S) (Anchukaitis et al., 2017) with the ensemble distributions of the four MPI-ESM ensembles (10 members). Anomalies are for the year 1810 wrt to 1800-1808 mean. The shading shows, for each grid point, the percentile range of the ensemble simulation where the reconstructed temperature falls. Green patches indicate that the reconstruction lies in the interquartile range of the simulations, hence is in good agreement with the ensemble. Bluish patches indicate that the reconstruction lies in the higher range of the ensemble, i.e, the majority of simulations are colder than the reconstructions. Reddish patches indicate that the reconstruction lies in the lower range of the ensemble, i.e., the majority of simulations are warmer than the reconstructions. White dots indicate where the reconstruction is an outlier with respect to the distribution of the simulation ensembles, i.e., where the absolute difference between reconstruction and simulation ensemble mean is greater than three times the median absolute deviation of the simulation ensemble.

North America. *Low* (Fig. 11c) and *nNHP* (Fig. 11d) are the experiments with the best agreement between the simulated and

the reconstructed surface temperature anomalies. *nNHP* is the experiment where, compared to the other model experiments, the reconstruction is in most locations within the interquartile range of the simulations, and which has the least number of locations for which the reconstruction is considered an outlier compared to the simulation ensemble. Overall, the N-TREND (S)





reconstruction is colder over Eastern Europe and West Russia compared to simulated surface temperature anomaly distribution in all four experiments, and warmer over the Eastern part of North America.

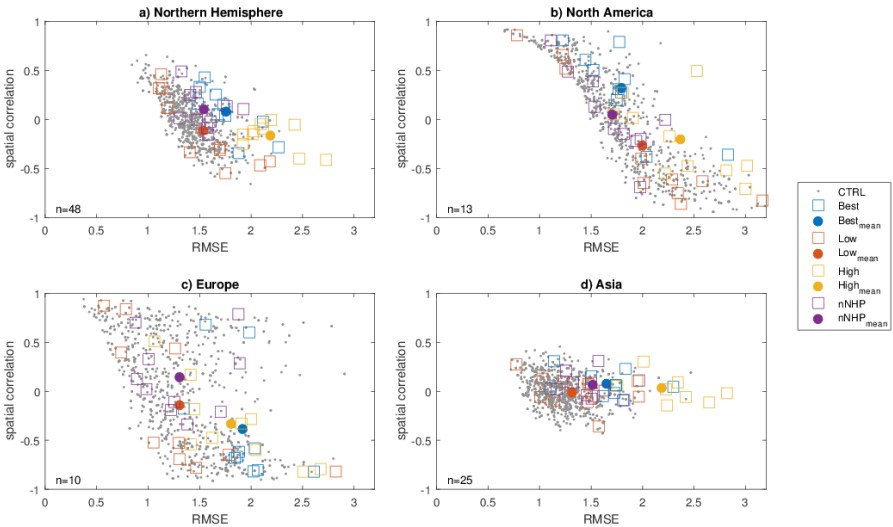

**Figure 12.** Scatterplots of Root Mean Square Error (RMSE) versus spatial correlation between simulated NH summer temperature and the N-TREND (S) reconstruction (Anchukaitis et al., 2017) for summer 1810 and different regions: a) whole Northern Hemisphere, b) North America (180-60°W), c) Europe (60°W-60°E), d) Asia (60-180°E). Individual model realizations are indicated by squares, the ensemble mean with a full dot; small grey dots are for 1000 random samples from the control period (1800 to 1808). Analysis is restricted to grid-points where proxy data are available (number of data used for each region reported in the respective panel).

Another method to compare reconstructed and simulated spatial patterns of temperatures anomalies in summer 1810 is shown in Fig. 12, which illustrates the spatial correlation and root-mean-square-error (RMSE) between the reconstruction and individual ensemble members of each experiment for the whole Northern Hemisphere and three equal-area sections of it. Similar metrics are calculated and shown for individual summers of an unforced control run, to illustrate the potential for the model to produce similar spatial structures as a result of natural variability. A perfect agreement between simulated and reconstructed data corresponds to spatial correlation of 1 and RMSE of 0, hence the best simulated representations of the

reconstructed anomalies are found close to the top-left corner of each panel. A perfect correlation could result from a simulation which had a bias compared to the reconstruction, while RMSE results from absolute differences due to spatial differences and biases. Over the entire NH (Fig. 12a) the scatter of all ensembles largely overlap with each other and with the control run, reflecting the effects of the relatively large internal climate variability. The *High* and *Best* ensembles yield ensemble-mean

metrics that are at the edge of the control run, with some realizations of the former ensemble being outside the spread of the control run for the NH and Asian regions. Both ensemble simulations are colder than the reconstruction (Fig. 11) for any year of the control run. The *Low* and *nNHP* ensemble compare best with the N-TREND (S) reconstruction according to this



analysis, especially as they yield the smallest RMSE values both regarding individual realizations and the ensemble-mean. The best correlations for the NH, in both the control and the forced experiments, are only 0.5 which reveals that the model does

not produce such a spatial pattern as we see in the reconstruction.

Model performance in terms of spatial correlation is especially interesting over North America (Fig. 12b), where a cold-warm zonal dipole is a major characteristic of the N-TREND (S) reconstruction but where also the proxy data quality is not optimal (Anchukaitis et al., 2017). The *Low*, *nNHP* and *Best* ensembles, as well as the control run, include realizations that yield high spatial correlations over North America. This suggests that such a continental anomaly is consistent with the variability

produced naturally by the model. The spread of correlation values over North America is wide, with many realizations showing strongly negative correlation with the reconstruction, and correlations of the ensemble means are small. There is therefore no evidence from the model that the spatial pattern of temperature anomalies over North America is a specific response to the volcanic aerosol forcing. The results for Europe (Fig. 12c) are similar to those of North America, with a handful of simulations from the *Low* and *nNHP* experiments showing highest correlation with the reconstruction, but no clear improvement of the

forced simulations in general compared to the control run. The range of correlations for the ensembles and the control run is smaller over Asia (Fig. 12d) than in the other considered regions, i.e., no realizations show strong positive (or negative) correlations with the reconstruction, as ensembles, except *nNHP*, yield too cold anomalies especially over Eastern Siberia that contrast the weak anomalies reconstructed there (Fig. 11). This is most likely related to a substantial data quality issue as the TR data, especially for central Asia, is solely based on TRW data (Wilson et al., 2016).

A comparison of simulated ensemble mean and observed near-surface air temperature anomalies over different European regions and New England is shown for NH summer and winter in Figs. 13,14, respectively, and with the individual ensemble members, which show a comparable variability to the observations in the appendix (Fig. S3-S10). Note that these figures neither account for the error in the observations nor for the representativity error (difference between a station and an areal average). For NH summer, the station data reflect the findings from the spatial comparison with the TR records (Fig. 11), i.e.,

most European station records indicate some cooling in summer 1810 (Fig. 13a-e), while the New England data shows no evidence of cooling in this year (Fig. 13f). Further, all of the stations show cooling in either 1812 or 1813, with many showing consecutive old summers until the Tambora eruption of 1815. In boreal summer 1810, the *nNHP* and *Low* simulations and station data are inconsistent over Northern Europe (Fig. 13c), where the observed cooling is larger than in the simulations whereas for *High* and *Best* the station data lie within the ensemble range. Surface air temperature over Western and Central

Europe seems to be not much affected by the 1809 eruption, neither in all model experiments nor in the station data (Fig. 13a,e). The simulated post volcanic cooling in summer 1810 is consistent with the station data over Southern and Eastern Europe in the model experiments with symmetric volcanic forcing (*Best*, *High*, *Low*) but *nNHP* shows slightly warm anomalies for summer 1810 (Fig. 13d,b). In contrast, *nNHP* is the only experiment which shows a similar trend to the New England stations, whereas the other experiments show stronger cooling there in 1810. Observed cooling after the Mt. Tambora eruption is matched quite

well by the model, except for Central Europe and Western Europe where the observed anomalies in 1816 are larger. Excellent agreement between model simulations and station data is found for the "year without summer" for New England. An interesting feature is the observed warming peak of 2 °C in summer 1811 over Central Europe, which is also found in one realization of

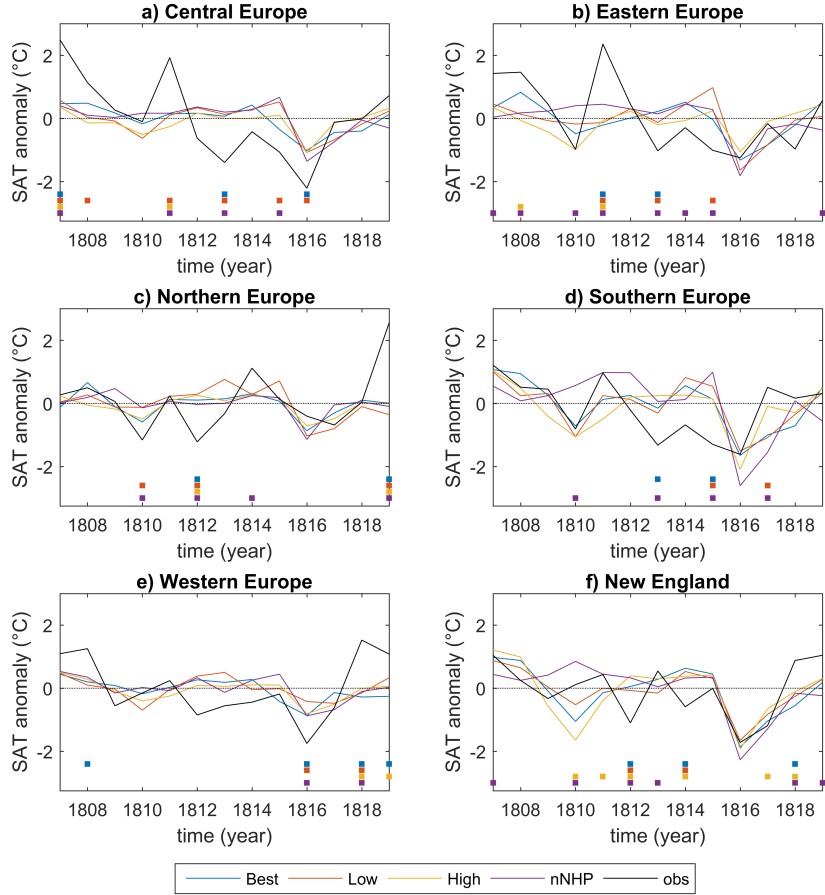

**Figure 13.** Seasonal mean near-surface NH summer temperature anomalies (°C) with station data averaged over different regions: a) Central Europe (46.1-52.5°N, 6-17.8°E), b) Eastern Europe (47-57°N, 18-32°E), c) Northern Europe (55-66°N, 10-31°E), d) Southern Europe (38-46°N, 7-13.5°E), e) Western Europe (48.5-56°N, 6°W-6°E) and f) New England (41-44°N, 73-69°W). The black line represents measurements, the colored lines the ensemble mean of the model simulations. The squares on the bottom of each panel indicate years where the measurement lies outside the simulation ensemble range (color code as for the ensemble mean). Anomalies are taken with respect to the years 1806-1820.

the *Best* experiment (Fig. S3) suggesting the influence of internal variability. For NH winter, both model and station data show higher variability than in NH summer (Fig. 14). Simulated and observed NH winter temperature anomalies agree quite well in the first three winters after the 1809 eruption. The only exception is New England (Fig. 14f) where, similar to NH summer (Fig. 13f), less agreement is found between the station data and the four experiments. The strong cooling signal of more than -2 °C which is found at Northern, Western and Central European stations in winter 1813 /1814 is not reproduced by the model





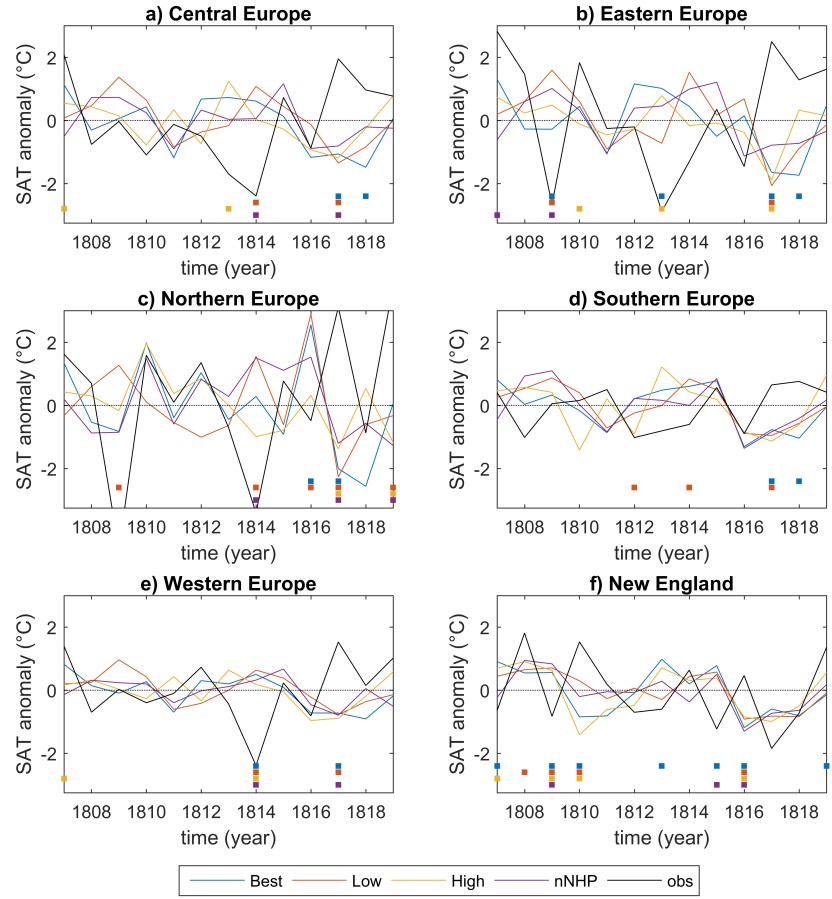

**Figure 14.** Same as Figure 13 but for seasonal mean near-surface NH winter temperature anomalies (°C). The year denotes the February month.

simulations. All experiments except *Low* produce positive post-volcanic winter temperature anomalies over Northern Europe with a warming signal in the first winter (1809/1810) after the 1809 eruption consistent with the station data.

## 5 Discussion

Significant surface cooling is found after the unidentified 1809 eruption in instrumental observations, proxy records and simulations, but the cooling is strongly spatially heterogeneous and is variable across the different data sources, especially across the NH extratropics.

Observed and simulated cooling in the tropics is considerable, with anomalies likely exceeding 0.5 °C. In general, the MPI-ESM simulations show stronger cooling over the tropical oceans than the reconstructed and observed temperatures (Fig. 7).



Over the tropical Indian ocean,the simulated SST anomalies between 1809 and 1820 for the *Low*, *Best* and *nNHP* experiments agree quite well with the marine ship measurements (Fig. 9). Furthermore, the ensemble mean cooling in *Low* is close to the reconstruction of D'Arrigo et al. (2006) over the tropical IndoPacific in 1809 although the reconstruction does not pick up the 1816 cooling expressed by all experiments (Fig. 7c). An even smaller SST variability range is also found for the Indian and

the West Pacific SST reconstructions when compared to the models (Fig. 8). Uncertainties related to the proxy measurements likely explain some of the mismatch, as they depend on several factors such as mixed sea surface temperature and salinity influences on the $\delta^{18}O$ measurements in coral archives as well as seasonal changes in the growth rate and other biological factors. The limited number of proxy records, which are not equally distributed in time and space, is also undoubtedly a relevant factor. Focusing on the tropical region, the comparison between simulations suggests that the most probable forcing

would be somewhere between *Low*, *nNHP* and *Best*.

Simulated ensemble mean NH extratropical summer temperature anomalies are mostly for all forcing scenarios within the reconstructed range but the agreement is strongly dependent on the choice of the year, spatial coverage and the individual forcing scenarios and reconstruction used. Ensemble mean temperature differences between the different MPI-ESM experiments or the individual reconstructions could be in the order of the forced signal (Fig. 10). The NH mean summer temperature response

to the 1809 eruption is smaller in TR reconstructions than in the simulations (except for *nNHP*), consistent with previous comparisons of TR reconstructions with climate models. Schneider et al. (2017), for example, compared TR reconstructions with PMIP3 last millennium simulations from different models including the MPI-ESM. They also found that on average most model simulations feature a more pronounced volcanic signal than proxy reconstructions. The spread of simulated NH temperature arising from uncertainties in the volcanic forcing estimate (VSSI) for the 1809 eruption is generally comparable to the

spread in the NH tree ring-based temperature reconstructions. It is therefore not possible to use one proxy result to strongly constrain the other, as we might if one distribution was much smaller than the other or if the overlap between the two was small. On the other hand, the disagreement between observed and simulated tropical temperatures in the *High* ensemble would suggest that VSSI estimates on the upper end of our estimated range are not likely and qualitatively, we might estimate that the most likely VSSI emissions for 1809 are between around 12 and 19 Tg S.

Instrumental records in Europe fit well with simulated temperatures using all forcing scenarios for both winter and summer seasons (Figs. 13,14). In boreal winter, observations from most stations (except New England and Eastern Europe) lie within the ensemble range of all four experiments within the 1st two post volcanic winters. Internal variability in NH winter is high and can overwhelm volcanic forcing uncertainties as pointed out by Zanchettin et al. (2019) for the 1st winter after the Mt. Tambora eruption. The spread of continental winter responses generated by the initial conditions for the unidentified 1809 eruption is

expected to be larger than for the Mt. Tambora eruption, due to the lower signal to noise ratio. Ultimately, comparison with early instrumental data must be made cautiously. Although regular meteorological measurements have been performed across Europe since the 17th and 18th centuries (Brönnimann et al., 2019b), continuous temperature measurements in the 1810s are rare due to the Napoleonic wars and there is a significant lack of regular temperature time series from other parts of the world for this period.



Spatial correlations over the NH between the simulations and the NTREND (S) temperature reconstruction for NH sum-
mer 1810 are overall low. None of the anomalous patterns produced by the MPI-ESM, either under unperturbed conditions or
volcanic-forcing scenarios, reproduces the amplitude and structure of the reconstructed hemispheric temperature anomaly pat-
tern particularly well in summer 1810 suggesting that the mismatch is most likely not attributable to the volcanic forcing. This
possibly stems from the heterogeneous reconstructed anomalies that lack a clear spatial structure and provide a "noisy" target

to the simulations. The quite unusual temperature pattern in N-TREND (S), with cold temperatures over Europe and Alaska
and warm temperatures over Eastern NA and Northern Siberia could be a potential cause for this. This heterogeneous spatial
pattern in the tree-ring data is also reflected in the dampened cooling seen in the NH mean composite reconstructions (Fig.
10). Station observations seem to support this anomaly pattern by showing cooling only over Northern and Eastern Europe.
Surface temperature reconstructions from an almost independent MXD network (Briffa et al., 2002b) indicate also positive

temperature anomalies over East Canada, north of 55°N and East of 95°W but also regional cooling. Briffa et al. used MXD
data only, which capture the short-time cooling after volcanic eruptions more reliably than RW based reconstructions which
are influenced by biological persistence (e.g., Anchukaitis et al., 2012; Esper et al., 2015; Lücke et al., 2019). In N-TREND (S)
a combination of MXD, RW and BI records are used and the prediction skill is spatially very variable, particularly over North
America (Anchukaitis et al., 2017). The distinct East-West dipole pattern in the reconstruction could therefore be partly caused

by a data quality issue as is certainly the case for Central Asia which is driven predominantly by tree ring-width data. One
possibility is that 1810 was a dynamically unusual but not unlikely year (Fig. 12) in terms of NH atmospheric circulation that
led to warming over Eastern NA which is not reflected in the simulated ensemble mean. While a few individual simulations
produced a dipole temperature anomaly pattern over North America similar to that of the NTREND (S) reconstruction when
forced with volcanic aerosol, many other simulations did not, and the ensemble mean correlation was close to zero for all

forced ensembles. We can only conclude that in these simulations, the volcanic aerosol forcing did not lead to or increase the
likelihood of the dipole North American pattern.

    Overall, our model-data intercomparison suggests that the mid-to-low range of the sulfate estimates for the unidentified 1809
eruption as the most realistic. However, the range of uncertainty is large, as observed and reconstructed data are sparse and cover
a wide post-volcanic temperature anomaly range. Uncertainties in volcanic forcing are often discussed as one of the main causes

for the discrepancy between simulated and reconstructed temperature response due to biases in magnitude and structure (e.g.,
Timmreck et al., 2009; Yang et al., 2019; Zhu et al., 2020). As discussed above, the spread of the different forcing ensembles
is in the same range as the different tree ring reconstructions, hence it is difficult to reduce the forcing estimates. Nevertheless,
our work does suggest that the response of large-scale NH climate indexes (Fig. 6) is sensitive not only to the magnitude of
the volcanic forcing, but also its meridional structure, with, e.g., significant differences in the summer NPI response for the

*Best* and *nHNP* ensembles. This would support the idea that the structure of forcing is important in influencing circulation
changes (e.g., Toohey et al., 2019). An asymmetric volcanic forcing scenario with less or no NH extratropical forcing, as in
*nHNP*, provides one way in which tropical temperatures could respond strongly to the 1809 eruption, while the NH temperature
response could be more muted and prolonged, as suggested by some TR reconstructions. Hence, an asymmetric hemispheric





forcing similar to that reconstructed for the March 1963 Agung eruption (Fig. S1) with strong tropical forcing but relatively

weak (and/or short-lived) NH extratropical forcing, appears to be a possible scenario for the 1809 eruption.

One final caveat to our simulation-reconstruction comparative assessment stems from the idealized nature of our experiments: our simulations are branched from a multi-centennial preindustrial control run and use only volcanic aerosol forcing; they are not transient simulations with realistic full forcing (including also variations in solar and land-use forcing, etc.). Neglecting the reduced solar irradiance in the early 19th century could, for example, be one of the factors why our model

simulations do not show the slight cooling trend in summer surface temperature between 1810 and 1815 indicated by the NH TR data. While this more idealized experimental set up has the advantage of better isolating the influence of volcanic forcing strength and structure alone, it complicates the comparison with observations and proxy reconstructions. Early 19th century ensemble simulations starting from a climate state with a more realistic past natural forcing, as planned for the CMIP6 VolMIP/PMIP "volc-cluster-mill"experiments (Zanchettin et al., 2019; Jungclaus et al., 2017), will allow for a more direct

comparison with early instrumental data and paleoclimate reconstructions. Furthermore, they will also help to disentangle the processes which are specific to the MPI-ESM model and which need discussion in a broader context.

## 6 Conclusions

We used the MPI Earth system model to study the climatic impacts of the unidentified 1809 eruption, the 6th largest volcanic eruption of the last millennium. Our aim was to address the question of whether or not the short-term climate response to the

1809 eruption noted in observations and proxy-based climate reconstructions is compatible with a range of volcanic forcing estimates constructed based on estimated uncertainties in volcanic stratospheric injection from ice core records. We demonstrated that the model simulations of the climate impacts of the 1809 eruption show generally good agreement with many of the available large-scale temperature reconstructions and early instrumental records. Assuming the model climate sensitivity is correct, this result implies that the eVolv2k VSSI estimate for the 1809 eruption is accurate within its reported uncertainty. Model-data

intercomparison of tropical temperature anomalies, where the impact of direct radiative forcing is highlighted, suggests that the most likely forcing is somewhere between the *Low* and the *Best* scenarios with an estimated sulfur emission between 12 -19 Tg S. However, the limited data availability for this time period for the tropical region poses a caveat to this conclusion. Simulated NH-average summer temperature anomalies are for all forcing scenarios within the observational and reconstructed range. The long-lasting cooling trend detected in observed and reconstructed NH-average summer temperatures between 1810

and 1815, i.e., until the Mt. Tambora eruption, is not detected in the model simulations except, partly, in *nNHP*. The reason(s) for such a cooling trend remains thus unclear and needs to be further explored. The spatial correlation between simulated and reconstructed near-surface air temperatures over the NH in summer 1810 is low. Tree-ring based gridded reconstructions of NH extratropical summer SAT show a highly heterogeneous anomaly pattern in 1810 with extensive regions of cold and warm anomalies, with an east-west warm-cold dipole over North America, and variable temperature anomalies over Asia, which

leads to a dampened NH-average cooling for this year (Fig. 10). The pattern over NA is consistent with the limited available station observations from New England, an almost independent MXD based summer temperature reconstruction (Briffa



et al., 2002b), and is reproduced by some forced simulations and years from the control run, suggesting it is a plausible result of natural variability. In contrast, the reconstructed temperature pattern over Asia is not produced by any model simulation, forced or control. Possible explanations for this model-data disagreement include deficiencies and uncertainties regarding both

tools. In particular, the dendrochronological network remains weakly constrained over the central Asian region, especially with respect to available MXD or density-like parameters, and instrumental observations are sparse, especially prior to the 1950s making calibration and validation difficult (Cook et al., 2013). Anchukaitis et al. (2017) clearly detailed poor reconstruction validation for many regions across North America as well as Northeast and central Asia. These regions must be targeted in ongoing proxy network development to update and develop new MXD or density-like parameter data-sets which are proven to

be superior proxies of volcanically forced summer cooling (e.g., Anchukaitis et al., 2012; D'Arrigo et al., 2013; Esper et al., 2015; Lücke et al., 2019). Concerning the simulations, beyond model deficiencies that may bias the response, our study demonstrates that choices regarding both volcanic forcing strength and spatial structure can similarly affect reconstruction-simulation comparative assessments. Specifically, it was shown that for the muted response to the 1809 eruption in the NH extratropics, agreement between reconstructions and simulations improves by weakening the magnitude of the eruption and, alternatively,

by preventing the volcanic aerosol to spread over the Northern Hemisphere. The forcing estimate of 1809, which is currently based on ice core data, can only be improved by modelling experiments to narrow down the uncertainty range. This will be facilitated if further information on location and eruption season is identified. At a broad-scale, latitude can be identified using S isotopes (Burke et al., 2019) which could at least help constrain whether *nNHP* experiments are potentially valid to improve the characterization of the climatic imprint of the 1809 eruption. Finally, it is clear that an increase in reconstruction accuracy

by improving spatial coverage, including also the southern hemisphere, is needed. Otherwise this eruption will remain most likely a mystery and a climatic cold case.





# 1    Appendix A Calculation of Indices

Definitions and indices are the same as described in Zanchettin et al. (2015). All indices are calculated on a monthly basis. The PNA index is applied to 500 hPa geopotential height (Z500) data. It is defined as $Z^*_{[15\check{~}25°N;180\check{~}220°E]}$ - $Z^*_{[40\check{~}50°N;180\check{~}220°E]}$ +

$Z^*_{[45\check{~}60°N;235\check{~}255°E]}$ - $Z^*_{[25\check{~}35°N;270\check{~}290°E]}$, where $Z^*$ denotes monthly mean and spatial averaged Z500 anomalies from the respective climatological value, with the suffix [x] for the region x. The NAO index is calculated based on the latitude–longitude two-box method from Stephenson et al. (2006) applied on Z500 data, i.e., as the pressure difference between spatial averages over 20–55°N; 90°W–60°E and 55–90°N; 90°W–60°E. The NPI is calculated using the definition from Trenberth and Hurrell (1994) applied to sea level pressure (SLP) data. The index is computed as spatial SLP averaged over 30–65°N;

160–220°E, so that positive phases of the index indicate a weaker-than-normal Aleutian low and the opposite holds for the negative phases. The SOI is defined as the difference between the average SLP over the domains 20–15°S; 147–152°W and 15–10°S; 128.5–133.5°E.

# 2    Appendix Station data

**Table A1.** Temperature observations from land stations used in this study. n = number of available value in the 1806-1820 period. Source: Dove = digitized from the collection of Friedrich Wilhelm Dove (Dove, 1838, 1839, 1842,1845), NOAA = digitized from images held at the NOAA archive, DWD = Germane Weather Service, GHCN = Global Historical Climatology Network (Lawrimore et al., 2011), ISTI = International Surface Temperature Initiative (Rennie et al., 2014).

| Source | Station | Start year | End year | lat | lon | N |
|--------|---------|------------|----------|-----|-----|---|
| Dove | London | 1787 | 1838 | 51.51 | -0.13 | 180 |
| Dove | Arnhem | 1790 | 1818 | 51.99 | 5.90 | 156 |
| Dove | Penzance | 1807 | 1827 | 50.14 | -5.49 | 168 |
| Dove | Dumfernline | 1805 | 1824 | 56.08 | -3.43 | 180 |
| Dove | Carlisle | 1801 | 1824 | 52.13 | -0.50 | 180 |
| NOAA | Brunswick | 1807 | 1820 | 43.92 | -69.94 | 142 |
| DWD | Berlin | 1719 | 2018 | 52.45 | 13.30 | 180 |
| DWD | Hohenpeissenberg | 1781 | 2018 | 47.80 | 11.01 | 156 |
| DWD | München | 1801 | 1953 | 48.14 | 11.60 | 180 |
| DWD | Stuttgart | 1792 | 1984 | 48.77 | 9.18 | 180 |
| DWD | Trier | 1788 | 2018 | 49.73 | 6.61 | 132 |
| GHCN | Mikolai | 1808 | 1875 | 47.03 | 31.95 | 96 |
| GHCN | New Haven | 1781 | 1970 | 41.30 | -72.90 | 180 |
| GHCN | Boston | 1743 | 2010 | 42.37 | -71.03 | 120 |
| GHCN | Kremsmuenster | 1767 | 2010 | 48.05 | 14.13 | 180 |
| GHCN | Innsbruck | 1777 | 1991 | 47.30 | 11.40 | 180 |





| Source | Station | Start year | End year | lat | lon | N |
|--------|---------|-----------|----------|-----|-----|---|
| GHCN | Praha | 1771 | 1981 | 50.10 | 14.25 | 180 |
| GHCN | Leobschutz | 1805 | 1849 | 50.20 | 17.80 | 180 |
| GHCN | Kobenhavn | 1768 | 1980 | 55.68 | 12.55 | 180 |
| GHCN | Torneo | 1801 | 1832 | 66.40 | 23.80 | 180 |
| GHCN | Woro | 1800 | 1824 | 63.20 | 22.00 | 180 |
| GHCN | Montdidier | 1784 | 1869 | 49.70 | 2.60 | 180 |
| GHCN | Paris | 1757 | 1980 | 48.80 | 2.50 | 180 |
| GHCN | Chalons | 1806 | 1848 | 48.90 | 4.40 | 180 |
| GHCN | Strasbourg | 1801 | 1991 | 48.55 | 7.63 | 180 |
| GHCN | Nice | 1806 | 1991 | 43.65 | 7.20 | 180 |
| GHCN | Karlsruhe | 1779 | 1930 | 49.03 | 8.37 | 180 |
| GHCN | Regensburg | 1773 | 2010 | 49.05 | 12.10 | 180 |
| GHCN | Budapest | 1780 | 1991 | 47.52 | 19.03 | 180 |
| GHCN | Udine | 1803 | 1991 | 46.00 | 13.10 | 180 |
| GHCN | Torino Casell | 1753 | 1981 | 45.22 | 7.65 | 180 |
| GHCN | Milano | 1763 | 1987 | 45.43 | 9.28 | 180 |
| GHCN | Verona | 1788 | 1991 | 45.38 | 10.87 | 144 |
| GHCN | Padova | 1780 | 1827 | 45.40 | 11.85 | 167 |
| GHCN | Bologna | 1808 | 1981 | 44.53 | 11.30 | 156 |
| GHCN | Palermo | 1791 | 1868 | 38.10 | 13.40 | 179 |
| GHCN | Riga | 1795 | 1989 | 56.97 | 24.05 | 102 |
| GHCN | Vilnius | 1777 | 1981 | 54.63 | 25.10 | 180 |
| GHCN | Trondheim | 1761 | 1981 | 63.40 | 10.50 | 180 |
| GHCN | Gdansk | 1807 | 1984 | 54.40 | 18.60 | 168 |
| GHCN | Warszawa | 1779 | 1991 | 52.17 | 20.97 | 180 |
| GHCN | Wroclaw | 1792 | 1991 | 51.10 | 16.88 | 180 |
| GHCN | St.Petersburg | 1743 | 1991 | 59.97 | 30.30 | 180 |
| GHCN | Stockholm | 1756 | 1980 | 59.33 | 18.05 | 180 |
| GHCN | Geneve | 1753 | 1991 | 46.25 | 6.13 | 180 |
| GHCN | Gordon Castle | 1781 | 1975 | 57.60 | -3.10 | 180 |
| GHCN | Edinburgh | 1764 | 1960 | 55.90 | -3.20 | 180 |
| GHCN | Manchester | 1794 | 1991 | 53.35 | -2.27 | 180 |
| GHCN | Greenwich | 1763 | 1969 | 51.50 | 0.00 | 180 |
| ISTI | Torino | 1760 | 2009 | 45.07 | 7.67 | 180 |
| ISTI | Basel | 1760 | 2010 | 47.54 | 7.58 | 180 |
| ISTI | Bern | 1777 | 2010 | 46.99 | 7.46 | 180 |
| KNMI | DeBilt | 1706 | 2018 | 52.10 | 5.19 | 180 |



**Table A2.** Overview of the different stations which are used in the compilation of the regional and seasonal station mean

| Region | Latitude/ longitude | Number of stations | Stations |
|---|---|---|---|
| Central Europe | 46.1-52.5°N,6-17.8°E | 17 | Geneve, Trier, Bern, Basel, Strasbourg, Karlsruhe, Stuttgart, Hohenpeißenberg, Innsbruck, München, Regensburg, Berlin, Kremsmuenster, Praha, Wroclaw, Leobschutz, Kobenhavn |
| Eastern Europe | 47°N-57°N, 18-32°E | 6 | Gdansk, Budapest, Warszawa, Riga, Vilnius, Mikolai |
| Northern Europe | 55-66°N, 10 -31°E | 5 | Trondheim, Stockholm, Woro, Torneo, St. Petersburg |
| Southern Europe | 38-46°N, 7- 13.5°E | 9 | Nice, Torino-Casell, Torino, Milano, Verona, Bologna, Padova, Palermo, Udine |
| Western Europe | 48.-56°N, 6 °W-6°O | 13 | Penzance, Dunfermline, Edinburgh, Gordon Castle, Carlisle, Manchester, London, Greenwich, Paris, Montdidier, Chalons, De Bilt, Arnhem |
| New England | 41-44°N, 73-69°W | 3 | New Haven, Boston, Brunswick |



*Code and data availability.* Primary data and scripts used in the analysis and other supplementary information that may be useful in
reproducing the author's work are archived by the Max Planck Institute for Meteorology and can be obtained by contacting publications@mpimet.mpg.de. Model results will be available under cera-www.dkrz.de soon. Proxi data can be obtained from NOAA/World Data
Service for Paleoclimatology archives https://www.ncdc.noaa.gov/paleo-search.

*Author contributions.* CT, MT and DZ designed the study, analysed the results and wrote the paper. CT performed the simulations and
coordinated the writing process. SB and RW provided data and contributed to the discussion and the writing process. EL compiled a new set
of the station data.

*Competing interests.* The authors declare that they have no conflict of interest

*Acknowledgements.* The authors thank Stephan Lorenz who gave valuable comments on an earlier version of the paper. Computations were
performed on the computer of the Deutsches Klima Rechenzentrum (DKRZ). The research has been supported by the Deutsche Forschungs-
gemeinschaft Research Unit VolImpact (FOR2820) within the project VolClim (TI 344/2-1, CT). SB acknowledges funding from the Swiss
National Science Foundation project WeaR (188701) and the European Commission through H2020 (ERC Grant PALAEO-RA 787574).
This work benefited from the participation of the authors in the Volcanic Impacts on Climate and Society (VICS) working group of the Past
Global Changes (PAGES) project.



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
