# Peer review of "The unidentified eruption of 1809: a climatic cold case"

_Climate of the Past, 2021_

## Author Response (AR1)

**Answer to Michael Sigl (Editor)**

We would like to thank Michael Sigl for his effort and his very constructive comments, which helped us to clarify an important aspect of the manuscript. In the following, we provide point-by-point replies. The editor comments are in blue and bold, answers are in black, cited text in italic and new or changed text is marked in red.

**Editor Decision: Publish subject to minor revisions (review by editor)** (06 May 2021) by Michael Sigl

**Comments to the Author:**

**Dear authors,**

**Your paper has received two reviews, and I would like to thank theme both for their work. Both reviewers agree in their assessment of the paper and acknowledge the high quality of the research and the manuscript. You have now responded to each of the review comments, and provided suggestions to improve the manuscript accordingly. In addition to the suggested edits made in response to the reviewer's comments and described in your responses, please also take the comments below into account for your revised manuscript.**

**General comment:**

**I admit I am biased, but I do miss a more nuanced discussion of the ice-core evidence in this paper. After all, ice cores are the principle witness in this whole case, if I may use your terminology. If it wasn't for the ice-cores, the climate effects in 1809/1810 probably would have been attributed to internal variability.**

**Ice cores not only constrain the timing, season and SO2 injection (though with uncertainties), but I believe they are also the only proxy that are able to eventually get this case moving again. The 1809 case of today is basically the same as the 1257 case ten years ago, which ultimately was solved by connecting the geochemistry of ash shards in ice cores with corresponding proximal fallout identified in the tropics around Samalas (Lavigne et al., 2013). Numerous other cases (Paektu, Eldgja, Ilopango, Okmok, Aniakchak) have been solved in the last 10 years owing to advancements in ice-core analyses techniques.**

**Even without a geochemistry match to a historic eruption, state-of-the-art ice-core records offer a wealth of insight into this specific case study which have not yet been fully employed. This must not be part of this paper, but at least the available data should be use as guidance to assess how plausible the four discussed forcing scenarios were. I agree with Reviewer 2 that the discussion of atmospheric circulation indices (L. 347-367, Fig. 6) are slightly disconnected from the rest of the paper, and I support your idea to move this into Supplementary Material. By creating extra space, you could show some actual ice-core records from Greenland and Antarctica (many of which are publically available). This will visualize that a nNHP scenario is indeed unlikely.**

**Volcanic forcing is reconstructed solely on the basis of volcanic sulfate aerosols locked in the ice sheets, and thus it is fully independent of any reconstructions and observations of past climate. Using agreement of observed/reconstructed versus simulated response (using one specific climate model) as criteria to judge the accuracy of the forcing is a possible approach. But also questionable, given the uncertainties in all entities. In this respect 1809 is not alone: There are other eruptions for which the climate response deviates (in both directions) from our expectations based on the forcing (e.g. 1600, 1453, 1257), some of which are discussed in more details in this Special Issue.**

We agree that the nNHP scenario is rather inconsistent with the ice-core evidence for 1809, and we have revised the text of its introduction to emphasize this point. Still, we think there is value in testing

this idealized and extreme scenario, even if it is not entirely consistent with the ice-cores. Our revised text is included in our answer to a specific comment below.

A quick look suggests to us that there is nothing conspicuous about the deposition of the 1809 aerosol to Greenland that would obviously support the nNHP scenario. We now mention the typical character of the duration of the 1809 deposition in our introduction of the nNHP forcing, to emphasize the fact that this scenario is not supported by the ice-core records, to hopefully avoid any confusion. Nevertheless, we feel that a detailed analysis of the high resolution ice-core data is beyond the scope of the paper.

**Additional comments:**

**Title: I agree with the critique by reviewer 2, and agree with your suggested new title. It is basically a cold case because we lack information on the location source and the season of the eruption. As soon as (or if) these will be revealed by some "smoking gun" the case will be opened again.**

After some additional thoughts we would like to slightly change the title again by removing the "still" to *"The unidentified eruption of 1809: a climatic cold case"* .

Our title will then have a double meaning:

1. 1809 has been a "cold case" because it doesn't get as much attention as other eruptions, like Tambora

2. 1809 is a "cold case" because we show that it produced cooling (even if it's not so clear in the NH tree rings)

**L. 2: Eruption magnitude (M) is already used by volcanologists with a different meaning than you have in mind.**

We have revised the sentence : *"Even though the eruption ranks as the 3rd largest since 1500 with a sulfur emission strength estimated to be two times that of the 1991 eruption of Pinatubo, not much is known of it from historic sources."*

**L. 43: This is imprecise: Timing and magnitude of SO2 injection are uncertain, the location is unknown.**

We have revised the sentence:

*"While the location and the magnitude of the 1809 eruption are unknown, it's exact timing is also uncertain."*

**L. 44: Most of the high-resolution ice-core studies have been performed over the past 10 years providing continuous sulfur and sulfate records from many ice-core sites in Greenland and Antarctica. These provide dates for the onset of sulfate deposition, the time duration of deposition on both polar ice sheets which are important to discuss in the light of the plausibility of your nNHP scenario.**

Point well taken. We have modified the text here to include support from the other high-resolution records:

*"A detailed analysis of high-resolution ice-core records points to an eruption in February 1809 ± 4 months (Cole-Dai, 2010), which is consistent with the timing implied by other high-resolution ice-core records (Sigl et al., 2013, 2015; Plummer et al., 2012)."*

Concerning the duration of the 1809 deposition signal, we include this point in our revised text as detailed in our response to the next comment.

**L. 124-131: I disagree: interhemispheric symmetry is the most likely scenario given the current knowledge. All ice-cores in Greenland and Antarctica show consistently high volcanic sulfate levels recorded at least between late 1809 to early 1811, and with S-isotopes confirming the stratospheric nature of sulfate. I don't see any evidence that would suggest that the lifetime was short or that aerosol deposition was restricted to winter. I think it is interesting to study such an extreme member (as an analogue for Agung 1963), though it is unlikely to realistically represent the 1809 event.**

We have revised our original text:

*"In fact, given the current knowledge, an interhemispheric symmetric forcing is only one of several realistic possibilities for the 1809 eruption. In recognition of these additional sources of uncertainty (and motivated in part by the apparent muted temperature response to the 1809 eruption in NH tree ring records) we constructed a fourth forcing set, which is identical to the "Best" forcing in the tropics and in the Southern Hemisphere (SH), but has the aerosol mass in the NH extratropics completely removed. Such forcing may seem unrealistic given the significant sulfur deposition to Greenland from the 1809 eruption, but we surmise that weak NH radiative forcing (relative to the Greenland sulfur signal) could result if the aerosol cloud had a short lifetime in the NH, especially if transport and removal of aerosol occurred primarily in the boreal winter when solar insolation is small in the NH. This forcing scenario, which we call "no-NH plume" or "nNHP" in the following, should be interpreted as a rather extreme "end-member" in terms of NH forcing."*

as follows :

*"Motivated in large part by the post-1809 surface temperature anomalies to be discussed below, which include strong cooling in the tropics and a muted NH mean temperature signal, we constructed a fourth forcing set, which is identical to the "Best" forcing in the tropics and in the Southern Hemisphere (SH), but has the aerosol mass in the NH extratropics completely removed, creating a strongly asymmetric forcing structure. This forcing scenario, which we call "no-NH plume" or "nNHP" in the following, should be interpreted as a rather extreme "end-member" in terms of NH forcing.*

*The lack of aerosol for the NH in this constructed forcing is clearly inconsistent with the polar ice-core records of sulfate deposition from the 1809 eruption, which suggests roughly equal deposition between Greenland and Antarctica over a similar duration to other typical tropical eruptions, indicating a long-lasting and global aerosol spread. Due to uncertainties in the conversion of ice-core sulfate to hemispheric aerosol burden and radiative forcing (Toohey and Sigl, 2017, Marshall et al., 2020), it is not impossible that the radiative forcing from the 1809 eruption aerosol was characterized by some degree of hemispheric asymmetry in reality. Still, the nNHP forcing presented here should be interpreted as a rather unlikely scenario for the 1809 eruption. We explore here the impact of this forcing scenario as an extreme, idealized form of hemispheric asymmetry that might conceivably play some role in the response to the 1809 eruption, and is directly applicable to "unipolar" tropical eruptions like Agung (1963) and El Chichon (1982)."*

**L. 443: nNHP shows the smallest anomalies not Low**

*Low* shows the smallest temperature anomalies with respect to the mean of the years 1800 to 1808. To clarify this, we have revised the sentence:

*"Between 1813 and 1815, nNHP reveals similar temperature anomalies to Best and High, while Low shows less cooling than all other experiments, which even disappears before the onset of the Tambora eruption."*

**L. 497: Typo: old instead of cold**

Corrected

**L. 530: You could also state that the strong tropical cooling provides additional support that the source of the 1809 eruption is likely in the tropics (as was expected on the basis of the ice cores).**

*This is a good suggestion and worth to consider but we prefer to add the statement at the end of the sentence in line 254 instead of line 530:*

*"However, the cooling is again found to be of comparable magnitude after the 1809 and 1815 eruptions, and therefore hints to a tropical location of the 1809 eruption in agreement with the ice-core data."*

**L. 586-590: A good hypothesis should be able to explain all observations. The asymmetric forcing hypothesis fails to explain that about 60% as much sulfate as Tambora were deposited (Cole-Dai et al., 2009) in Greenland between 1809-11, which cannot be reconciled with an absence of a plume in the Northern hemisphere extra-tropics.**

*text at present:*

*"An asymmetric volcanic forcing scenario with less or no NH extratropical forcing, as in nNHP, provides one way in which tropical temperatures could respond strongly to the 1809 eruption, while the NH temperature response could be more muted and prolonged, as suggested by some TR reconstructions. Hence, an asymmetric hemispheric forcing similar to that reconstructed for the March 1963 Agung eruption (Fig. S1) with strong tropical forcing but relatively weak (and/or short-lived) NH extratropical forcing, appears to be a possible scenario for the 1809 eruption."*

*suggested edit:*

*"An asymmetric volcanic forcing scenario with less or no NH extratropical forcing, as in nNHP, provides one way in which tropical temperatures could respond strongly to a tropical eruption, while the NH temperature response could be more muted and prolonged, as suggested by some tree-ring reconstructions for 1809. This scenario is however at odds with the ice-core records, which show comparable deposition to Antarctica and Greenland suggesting a global aerosol spread. Without a viable explanation for a mismatch between ice-core records and hemispheric radiative forcing, extreme hemispheric asymmetry (as in the nNHP scenario) must be concluded to be an unlikely reason for the muted NH mean temperature response to the 1809 eruption. "*

**L. 635-36: Why "currently"? I don't see any other means by which the forcing can be estimated any times soon.**

Correct, we have deleted "currently".

**L. 637-39: Sulfur isotopes do not help to identify the latitude. Performed at low resolution (as in Cole-Dai et al., 2009 for Greenland and Gautier et al., 2019 for Antarctica for 1809) they provide information if the bulk of the sulfate was formed in the stratosphere (i.e. above the tropopause). Performed at high (sub-annual) resolution they may have the potential to disentangle additional source eruption parameters (plume height; presence of tropospheric sulfate) and eventually discriminate volcanic sulfate from distinct separate eruptions, but these measurements are still very scarce.**

This is correct. We have implicitly assumed that if we got a clear stratospheric sulfur isotope signal the eruption was located in the tropics. We have to admit that this assumption is oversimplified . We have revised the paragraph and removed the link to the isotopes.

*"This will be facilitated if further information on location and eruption season is identified. Recent advances in ice-core analysis can help to identify the source volcano of large historic eruptions by combining proximal tephra fallout with the geochemical analysis of ash shards in ice-cores as done for example for the 1257 Samalas eruption (Lavigne et al., 2013), the Changbaishan eruption in 946 (Oppenheimer et al., 2017), the 431 CE Tierra Blanca Joven eruption of Ilopango (Smith et al. 2020), or*

*the 1477 CE eruption of the Icelandic Veiðivötn–Bárðarbunga volcanic system (Abbott al., 2021, this issue).”*

**L. 639: climate reconstruction**

Changed to:

*“Finally, it is clear that improving temperature reconstruction accuracy by adding additional tree-ring density records in poorly replicated regions, including also the southern hemisphere, is needed. “*

**Additional changes**

In the light of the paper revisions we also have revised the end of the paper to:

From:

*“Finally, it is clear that an increase in reconstruction accuracy by improving spatial coverage, including also the southern hemisphere, is needed. Otherwise this eruption will remain most likely a mystery and a climatic cold case.”*

to:

*“Finally, it is clear that an increase in temperature reconstruction accuracy by improving spatial coverage, including also the southern hemisphere, is needed.*

*In summary, analysis of multiple climate records shows that the eruption of 1809 was certainly a “cold case” in the sense that it produced large-scale cooling, and that cooling is consistent with the range of sulfur emission and radiative forcing estimates deduced from ice core records. In terms of attributing the eruption to a source volcano, or explaining the spatially heterogeneous and temporally delayed cooling in the NH suggested by tree ring networks, the investigation remains open. “*

**References:**
**Gautier, E., Savarino, J., Hoek, J., Erbland, J., Caillon, N., Hattori, S., Yoshida, N., Albalat, E., Albarede, F., and Farquhar, J.: 2600-years of stratospheric volcanism through sulfate isotopes, Nat Commun, 10, 2019.**
**Lavigne, F., Degeai, J. P., Komorowski, J. C., Guillet, S., Robert, V., Lahitte, P., Oppenheimer, C., Stoffel, M., Vidal, C. M., Surono, Pratomo, I., Wassmer, P., Hajdas, I., Hadmoko, D. S., and De Belizal, E.: Source of the great A.D. 1257 mystery eruption unveiled, Samalas volcano, Rinjani Volcanic Complex, Indonesia, P Natl Acad Sci USA, 110, 16742-16747, 2013.**

In addition:
Abbott, P. M., Plunkett, G., Corona, C., Chellman, N. J., McConnell, J. R., Pilcher, J. R., Stoffel, M., and Sigl, M.: Cryptotephra from the Icelandic Veiðivötn 1477 CE eruption in a Greenland ice core: confirming the dating of volcanic events in the 1450s CE and assessing the eruption's climatic impact, Clim. Past, 17, 565–585, https://doi.org/10.5194/cp-17-565-2021, 2021.
Marshall, L. R., Smith, C. J., Forster, P. M., Aubry, T. J., Andrews, T. and Schmidt, A.: Large variations in volcanic aerosol forcing efficiency due to eruption source parameters and rapid adjustments, Geophys. Res. Lett., 47(19), doi:10.1029/2020gl090241, 2020.
Oppenheimer, C., Wacker, L., Xu, J., Galván, J. D., Stoffel, M., Guillet, S., Corona, C., Sigl, M., Di Cosmo, N., Hajdas, I., et al.: Multi-proxydating the 'Millennium Eruption'of Changbaishan to late 946 CE, Quaternary Science Reviews, 158, 164–171, 2017.
Plummer, C. T., Curran, M. A. J., van Ommen, T. D., Rasmussen,S. O., Moy, A. D., Vance, T. R., Clausen, H. B., Vinther, B.M., and Mayewski, P. A.: An independently dated 2000-yr vol-canic record from Law Dome, East Antarctica, including a newperspective on the dating of the 1450s CE eruption of Kuwae,Vanuatu, Clim. Past, 8, 1929–1940, https://doi.org/10.5194/cp-8-1929-2012, 2012.
Sigl, M., McConnell, J. R., Layman, L., Maselli, O., McGwire, K.,Pasteris, D., Dahl-Jensen, D., Steffensen, J. P., Vinther, B., Ed-wards, R., Mulvaney, R., and Kipfstuhl, S.: A new bipolar icecore record of volcanism from WAIS Divide and NEEM and im-plications for climate forcing of the last 2000 years, J. Geophys.Res., 118, 1151–1169, https://doi.org/10.1029/2012JD018603,2013.

Sigl, M., Winstrup, M., McConnell, J. R., Welten, K. C., Plun-kett, G., Ludlow, F., Büntgen, U., Caffee, M., Chellman, N.,Dahl-Jensen, D., Fischer, H., Kipfstuhl, S., Kostick, C., Maselli,O. J., Mekhaldi, F., Mulvaney, R., Muscheler, R., Pasteris, D.R., Pilcher, J. R., Salzer, M., Schüpbach, S., Steffensen, J. P.,Vinther, B. M., and Woodruff, T. E.: Timing and climate forc-ing of volcanic eruptions for the past 2,500 years, Nature, 523,543–549, https://doi.org/10.1038/nature14565, 2015

Smith, V. C., Costa, A., Aguirre-Díaz, G., Pedrazzi, D., Scifo, A., Plunkett, G., Poret, M., Tournigand, P.-Y., Miles, D., Dee, M. W., et al.:The magnitude and impact of the 431 CE Tierra Blanca Joven eruption of Ilopango, El Salvador, Proceedings of the National Academy ofSciences, 117, 26 061–26 068, 2020

Toohey, M. and Sigl, M.: Volcanic stratospheric sulphur injections and aerosol optical depth from 500 BCE to 1900 CE, Earth System Science Data, 9, 809–831, 2017.

**Answer to Reviewer 1**

We would like to thank Reviewer 1 for his/her effort and very constructive comments, which helped us a lot to improve the manuscript. In the following, we provide point-by-point replies. Reviewer comments are in blue and bold, answers are in black, cited text in italic and new or changed text is marked in red.

**Reviewer 1**

**Section 2.1.1: It might be worth introducing the parameterization scheme for the aerosol microphysical processes in MPI-ESM1.2-LR. As suggested in recent studies (e.g. LeGrande et al., 2016, Nat. Geosci.), some CMIP5-era climate models can produce overly strong volcanic cooling due to unrealistic aerosol microphysics. How is the scheme in MPI-ESM1.2-LR different?**

The simulation of aerosol microphysical processes is computationally very expensive therefore only a very few CMIP type models treat aerosol microphysical processes interactively. The MPI-ESM1.2 model does not consider aerosol microphysical processes explicitly. The radiative volcanic forcing is prescribed by monthly mean optical parameters (extinction, asymmetry factor, single scattering albedo) which are considered in the model's radiative scheme. The optical parameters are precalculated with the EVA volcanic forcing generator (Toohey et al., 2016), which we explain in detail in section 2.1.2. A multi-model comparison for the Tambora eruption of different aerosol microphysical models (Zanchettin et al., 2016; Clyne et al., 2020) revealed large differences between the different aerosol models and that the more idealized forcing approach with the EVA tool is within the multi-model range.

For clarification, we included now in the model description (2.1.1) the following sentences:

"*In ECHAM6.3 aerosol microphysical processes are not included. The radiative forcing of the volcanic aerosol is prescribed by monthly and zonal mean optical parameters which are generated with the EVA forcing generator (Toohey et al., 2016), see section 2.1.2.*"

To address the uncertainties in the microphysical processes we have added in the forcing description (2.1.2) the following sentences:

''*The applied volcanic forcing is compiled with the Easy Volcanic Aerosol (EVA) forcing generator (Toohey et al., 2016). EVA provides an analytic representation of volcanic stratospheric aerosol forcing, prescribing the aerosol's radiative properties and primary modes of their spatial and temporal variability. Although EVA represents an idealized forcing approach, its forcing estimates lie within the multi model range of global aerosol simulations for the Tambora eruption (e.g., Zanchettin et al., 2016; Clyne et al., 2020).*

**Eruption timing: In L107 it is mentioned that the 1809 eruption is set to occur on Jan 1st of 1809. Recent studies (e.g., Predybaylo et al., 2020, Commun. Earth & Environ) have suggested that the eruption timing may also affect the climate response, especially the ENSO response, due to different circulation conditions and ENSO phases. Since the SOI response is also assessed in Fig. 6, it might make this study more complete to add one extra experiment testing the sensitivity to the eruption timing.**

The Reviewer raised an important point about the influence of the eruption season on the climate response to large volcanic eruptions. This is of course of particular interest with respect to tropical hydroclimates and ENSO, which is discussed widely in the recent literature (e.g., Stevenson et al., 2017; Predybalo et al., 2017, 2020; Zhuo et al., 2021). However, this is beyond the central scope of this paper and we therefore refrain from doing an extra sensitivity study that would complicate our study and its main scientific conclusions. The eruption's season and its influence on tropical dynamics could be a topic of a separate study, specifically focused on a comparison of model data and tropical SST reconstructions for large historic eruptions. For such a study not only additional simulations for

different eruption seasons are of interest but also simulations accounting for different initial oceanic background states.

According to Reviewer 2 comments we will shift Figure 6 (which we have changed according to your suggestions) to the appendix (Figure S1) and shorten the text about the circulation results, see comment below. We have now included also the "eruption timing" as a critical point in our discussion:

*" ...with strong tropical forcing but relatively weak (and/or short-lived) NH extratropical forcing, appears to be a possible scenario for the 1809 eruption. A factor which might have an influence on the mismatch between model results and proxy data is the season of the eruption, which has an impact on the climate response to volcanic eruptions (e.g., Stevenson et al., 2017; Predybalo et al., 2020). We have not discussed this uncertainty for the 1809 eruption here as we chose January 1809 as the starting date following most studies that suggest that the eruption probably happened in boreal winter months across 1808 and 1809. Nevertheless, the timing of the eruption remains uncertain. Chenoweth (2001) for example dated the 1809 eruption back to March-June 1808 based on a sudden cooling in Malaysian land surface temperature data and a peak cooling of marine air temperature in 1809. To take into account the eruption's season may be important to fully characterize the range of climate response compatible with the 1809 eruption and should be addressed in further studies."*

**Fig. 2: It might be better to use green triangles or other readable colors and symbols to denote the location of the tree-ring proxies. Red can be misleading given that red also represents a high temperature anomaly. The colorbar may also be adjusted to drop the white color to differentiate the missing values.**

Thank you very much for your suggestion, we have revised Fig. 2 (shown below) and use pink dots for the tree ring locations. According to a comment from Reviewer 2 we also included a panel of the NH summer temperature anomalies of 1811 instead of 1816 in the figure, and shifted the 1816 NH summer panel to the supplements (Fig. S3).

[Figure]

**F**igure 2*: Observed and reconstructed temperature anomalies around the 1809 volcanic eruption. **a)** Reconstructed tropical (30°N–30°S, 34°E-70°W) sea surface temperature (TROP, D'Arrigo et al., 2009), measured tropical marine surface air temperatures from ship logs (EEIC, Brohan et al., 2012) and warm pool data (WPOOL, D'Arrigo et al. 2006). **b)** NH summer land temperatures from four tree-ring based reconstructions (Wilson et al,, 2016 (N-TREND (N)), Anchukaitis et al., 2017 (N-TREND (S)), Guillet et al., 2017 (NVOLC), Schneider et al., 2015 (SCH15)). **c)** Monthly mean NH winter and summer temperature*

*anomalies (°C) from 53 station data averaged over different European regions (Central Europe (CEUR: 46.1-52.5°N, 6-17.8°E), Eastern Europe (EEUR: 47-57°N, 18-32°E), Northern Europe (NEUR: 55-66°N, 10 -31° E), Southern Europe (38-46°N, 7-13.5°E), Western Europe (WEUR: 48.5-56°N, 6°W- 6°O) and New England (NENG: 41-44°N, 73- 69°W). **d-f)** Mean surface temperature anomalies (°C) for boreal summers of 1809 (d), 1810 (e) and 1811 (f) in NH TR data N-TREND (S) (Anchukaitis et al, 2017). Pink dots in panel d illustrate the location of the tree-ring proxies used in the N-TREND reconstruction. All anomalies are with respect to the 1800-1808 climatology.*

***Fig. 5 & 6d: It might be worth showing the relative sea surface temperature (RSST) (Khodri et al., 2017, Nat. Commun.) to highlight the impact of volcanic forcing on ENSO relative to tropical mean cooling in the supplementary information. It may or may not affect the results significantly, but either way it is a valuable assessment.***

We have now calculated the relative sea surface temperature according to Khodri et al. (2017) and included an additional plot showing the relative SST in the tropics in the supplements (Figure S5).

 We slightly revised the discussion of Figure 5:

*" High is the only experiment where a significant El Nino type anomaly is seen over the Pacific Ocean in boreal summer 1810, while in the other three experiments a slight but non-significant warming appears off the coast of South-American.* Looking to the relative SST anomalies as calculated after Khodri et al. (2017) an El Nino type anomaly is seen for all 4 scenarios in boreal summer 1810, while in winter 1809/1810 a significant warming anomaly appears in the central tropical Pacific in all experiments except Best (Fig. S5)."

[Figure]

***Figure S5:*** *Seasonal mean relative sea surface temperature anomalies in the tropical regions. Simulated ensemble-mean relative sea surface temperature anomalies for the 1ˢᵗ winter (1809/1810) and the second summer (1810) after the 1809 eruption for the four different MPI-ESM simulations. Shaded regions are significant at the 95% confidence level according to a t-test. Anomalies are calculated with respect to the period 1800-1808.*

**Fig. 6: If I am not misunderstanding, the volcanic forcing magnitude in Best, Low, and High experiments can be ranked as High > Best > Low according to Fig. 1, and there's no difference in their meridional structure. I am curious about the reason that the SOI response in the Low experiment seems to lie between that of the Best and High experiments during both winter and summer. Particularly, does the fact that the High and Best experiments show opposite signs during the summer indicate that the SOI response is actually more internally driven than externally forced? Similar doubts exist for other indices that the impact seems not following the same monotonic order of the magnitude of the forcing. It might be good to reorder the bars in the figures as pre, Low, Best, High to highlight the potential impact of magnitude, no matter exists or not.**

We have changed the order of the experiments in Figure 6. The revised figure is shown below. Indeed, the revised order better illustrates a possible link between the response of some of the indices and the magnitude of the applied forcing. This seems more evident in NAO, PNA, with noticeably opposite tendencies seen in the two seasons for both indices. The relation is less clear for NPI and SOI, which we interpret as being related to the different ENSO-related dynamics triggered by the different volcanic forcing. Following a comment by Reviewer 2, we have toned down the discussion about the circulation indices and put the amended Figure 6 as supplement in the revised manuscript, as a detailed analysis and discussion of the underlying dynamical aspects are beyond the scope of this paper and deserve a dedicated study. Specifically, we have moved appendix 1 to the supplement and revised the text as follows:

-lines 233-237 of the original manuscript are moved to the supplement

-the paragraph in lines 347-367 of the original manuscript is moved to the supplement, and replaced with a shortened version along the following lines: *"The substantial differences found in the post-eruption evolution of continental and subcontinental climates reflect the variety of climate responses produced by different combinations of internal climate variability and forcing structure. In this regard, post-eruption anomalies of selected dominant modes of large-scale atmospheric circulation in the Northern Hemisphere and the tropics, including the Pacific/North American pattern, the North Atlantic Oscillation, the North Pacific Index and the Southern Oscillation, yield a spread of responses within individual ensembles that is often as large as the range of pre-eruption variability. Further, response distributions generated by different forcings in some cases do not overlap (see Supplementary Figure S1). "*

[Figure]

**F*igure 6 (revised, now Figure S1).* Atmospheric circulation indices. Box-Whisker plots (median, 25th-75th and 5th-95th percentile ranges) of seasonal anomalies of circulation indices: a) North Atlantic Oscillation (NAO), b) Pacific/North American pattern (PNA), c) the North Pacific Index (NPI) and d) the Southern Oscillation Index (SOI) from the model experiments for the first post-eruption winter (1809/1810, DJF) and second post-eruption summer (1810, JJA) following the 1809 eruption. Pre-eruption (1800-1808) data, shown as grey Box-Whisker plots, are used to standardize the indices.*

**Fig. 7a and L200-204: In L200-204, it is mentioned that the authors accounted for the sparsity and irregularity in spatial and temporal sampling of the EEIC data, but it is unclear how good the performance of the processing is, and the authors still see overall dampened tropical SST anomalies in EEIC compared to model simulations in Fig. 7a. I was wondering what the comparison would look like if compare the mean of the model simulated SST anomalies over grid cells nearest to the locales of the EEIC logs to the mean of the original EEIC data. Similar strategy might be worth taking for**

**other comparisons if the observations/reconstructions are available over multiple sites instead of a processed regional mean.**

[Figure]

*Figure S6: Impact of limited sampling on the average tropical surface air temperature (SAT) anomalies in the MPI-ESM simulations. (top), EEIC and MPI-ESM tropical SAT anomalies, the solid colored lines show the overall simulated tropical (30°S-30°N) average values, while the dashed lines show the average of model points corresponding to the EEIC measurements. (bottom). Difference between the model values (Full sampling - EEIC sampling).*

Indeed, as suggested by the Reviewer, the sampling of the ship based measurements does introduce some uncertainty into the average values. Figure S6 (above) shows the results of two sampling strategies used on the model data: the solid lines show the average tropical SAT anomaly, while the dashed lines show the average of the model sampled at the point and time of each EEIC measurement. The difference between the two is shown in the bottom panel. This analysis shows that the sampling acts to diminish the temperature anomaly in 1809 compared to the full tropical average. The magnitude of this impact is around 10-30% in 1809. The impact in 1810, at the peak of the tropical cooling (in the model) is quite small. For the Tambora eruption, sampling impacts the mean differently in 1815 and 1816.

This comparison highlights, and provides a rough estimate of, the impact that limited sampling in the EEIC measurements has on estimating the tropical mean temperature anomaly. We have included the figure above in the supplemental information (Figure S5), and refer to it in the revised text. This uncertainty in the measurements does not however significantly impact the model overestimation of the tropical cooling in 1810.

*Added text (line 376): "When the model results are sampled at the locations and times of the EEIC measurements (Fig. S6), the mean negative temperature anomalies in 1809 are 10-30% smaller, with Best, High and nNHP experiments all producing anomalies similar to that of the EEIC measurements. For the 1810-1812 period, the sampling makes little difference compared to the full tropical average, with Best, High and nNHP experiments all showing larger negative temperature anomalies than the EEIC measurements.*

**Fig. 9: It seems that the strategy mentioned above is taken here in Fig. 9 as the model simulations are "similarly sampled". Perhaps can add an extra column for the visualization similar to Fig. 7a, comparing EEIC to ensemble means, but for two separated regions. Is it overall a better agreement in Fig. 9 than in Fig. 7a? If so, does it mean the sparsity of the EEIC logs is not well accounted for as mentioned in L200-204?**

The Reviewer is correct that the EEIC sampling of the model is used in Fig. 9. The inclusion of each ensemble member in this plot allows for a better appreciation of the variability in each ensemble of simulations, of which the sampling plays a part. As suggested, we propose to include a figure (S6, see above) displaying the impact of sampling on the tropical average of Figure 7a. in the supplements. As discussed above, while the sampling clearly makes an impact on the calculated averages, it does not appear to significantly change the interpretation of the comparison between the EEIC measurements and the model simulations.

*Fig. 13 & 14: What is the rationale that the anomalies are calculated with respect to the years 1806-1820 here instead of 1800-1808 as in previous figures? The decision will largely affect the model-data comparison on the response to volcanic forcing.*

We agree that this would be more consistent but unfortunately the station data are sparse and irregular and often only partly available between 1800- 1808. Hence, we decided to take into account the full period as a reference period only for this analysis.

**L527: a typesetting issue (delta-18-O)**

The typo is corrected.

**References:**

Anchukaitis, K.J., et al: Last millennium Northern Hemisphere summer temperatures from tree rings: Part II: spatially resolved reconstructions, Quaternary Science Reviews, 163, 1-22, doi: 10.1016/j.quascirev.2017.02.020, 2017.

Brohan, P., Allan, R., Freeman, E., Wheeler, D., Wilkinson, C., and Williamson, F.: Constraining the temperature history of the past millennium using early instrumental observations, Clim. Past, 8, 1551–1563, https://doi.org/10.5194/cp-8-1551-2012, 2012.

Chenoweth, M. Two major volcanic cooling episodes derived from global marine air temperature, AD 1807–1827 https://doi.org/10.1029/2000GL012648, 2001.

Clyne, M., et al.: Model physics and chemistry causing intermodel disagreement within the VolMIP-Tambora Interactive Stratospheric Aerosol ensemble, Atmos. Chem. Phys., 21, 3317–3343, 2021.

D'Arrigo, R. et al..: The reconstructed Indonesian warm pool sea surface temperatures from tree rings and corals: Linkages to Asian monsoon drought and El Niño–Southern Oscillation, Paleoceanography,21, PA3005, doi:10.1029/2005pa001256, 2006.

D'Arrigo, R., Wilson R., and Tudhope, A.: The impact of volcanic forcing on tropical temperatures during the past four centuries, Nat. Geosci., 2, 51–56, doi:10.1038/ngeo393, 2009.

Guillet, S., et al.: Climate response to the Samalas volcanic eruption in 1257 revealed by proxy records, Nature geoscience, 10, 123–128, 2017.

Khodri, M., et al.: Tropical explosive volcanic eruptions can trigger El Nino by cooling tropical Africa, Nat. Commun., 8(1), 778, doi:10.103 8/s41467-017-00755-6, 2017.

LeGrande, A. N., Tsigaridis, K., and Bauer, S. E.: Role of atmospheric chemistry in the climate impacts of stratospheric volcanic injections, Nat. Geosci., 9, 652–655,https://doi.org/10.1038/ngeo2771, 2016.

Predybaylo, E., G. L. Stenchikov, A. T. Wittenberg, and F. Zeng, 2017: Impacts of a pinatubo-size volcanic eruption on enso. J. Geophys. Res., 122, 925-947, https://doi.org/10.1002/2016jd025796.

Predybaylo, E., Stenchikov, G., Wittenberg, A. T., and Osipov, S.: El Niño/Southern Oscillation response to low-latitude volcanic eruptions depends on ocean pre-conditions and eruption timing. Communications Earth & Environment, 1(1), 1-13, 2020.

Stevenson, S., Fasullo, J. T., Otto-Bliesner, B. L., Tomas, R. A., and Gao, C.: Role of eruption season in reconciling model and proxy responses to tropical volcanism, P. Natl. Acad. Sci. USA, 114(8), 1822-1826, doi:10.1073/pnas.1612505114, 2017.

Schneider, L., Smerdon, J. E., Büntgen, U., Wilson, R. J. S.,Myglan, V. S., Kirdyanov, A. V., and Esper, J.: Revising mid-latitude summer temperatures back to A.D. 600 based on a latewood density network, Geophys. Res. Lett., 42, GL063956,https://doi.org/10.1002/2015gl063956, 2015.

Toohey, M., Stevens, B., Schmidt, H., and Timmreck, C.: Easy Volcanic Aerosol (EVA v1.0): an idealized forcing generator for climate simulations, Geosci. Model Dev., 9, 4049 – 4070, doi:10.5194/gmd-9-4049-2016, 2016.

Wilson, R., et al: Last millennium northern hemisphere summer temperatures from tree rings: Part I: The long term context, Quat. Sci. Rev., 134, 1–18, doi:10.1016/j.quascirev.2015.12.005, 2016.

Zanchettin, D., Khodri, M., Timmreck, C., Toohey, M., Schmidt, A., Gerber, E. P. et al.: The Model Intercomparison Project on the climatic response to Volcanic forcing (VolMIP): experimental design and forcing input data for CMIP6, Geosci. Model Dev., 9, 2701 – 2719, doi:10.5194/gmd-9-2701-2016, 2016.

Zhuo, Z., Kirchner, I., Pfahl, S., and Cubasch, U.: Climate impact of volcanic eruptions: the sensitivity to eruption season and latitude in MPI-ESM ensemble experiments, Atmos. Chem. Phys. Discuss. [preprint], https://doi.org/10.5194/acp-2021-260, in review, 2021.

**Answer to Oliver Bothe (Reviewer 2)**

We would like to thank Oliver Bothe for his effort and his very constructive comments, which helped us a lot to improve the manuscript. In the following, we provide point-by-point replies to his comments. The Reviewer's comments are in blue and bold, answers are in black, cited text in italic and new or changed text is marked in red.

**Major**

**That is: I do not believe that the manuscript does what the authors claim in the title. While I appreciate a catchy title, I think overselling is a net-minus. My understanding of the title is that the authors claim to show why 1809 remains a cold case. The title plays with the cold-case terminology from criminal investigations and procedural crime series on TV and streaming. However, from my point of view the manuscript does not show enough in that respect to claim this title. A positive reading is, the title already announces the failure to provide major new insights into location, seasonality, or strength of the eruption, a negative reading is that the title suggests large new insights, why we won't have much success in becoming more certain about this eruption. I don't think these large new insights are provided. The manuscript is in a sense incremental while also presenting some very valuable - and I think new - simulations and in addition supporting previous understanding about the eruption in 1809.**

We agree with the Reviewer that we do not provide enough new material in our study to justify the "why it remains" in the title. Nevertheless, our findings contribute to the recent knowledge of the unidentified 1809 eruption and support previous understanding (and lack thereof) of its climate impact. Hence, we propose to revise our title to: *"The unidentified eruption of 1809: a climatic cold case".*

**Minor**

**Line 19: The claim that the early 19th century is the coldest period of the last 500 years is based on a reference to Jungclaus et al. (2017). While I am not entirely sure whether this is meant to be a global statement or a hemispheric one, there have been a number of recent reconstruction efforts spanning the period of the last 500 years and beyond, which may or may not require to reassess or qualify the statement. That is, such an absolute statement requires assessing the newest evidence, which in this case may include, for example, Büntgen et al. (2020), PAGES2k Consortium (2019), and more.**

The reference to Jungclaus et al (2017) was related to the 2$^{nd}$ part of the sentence that the early 19$^{th}$ century is a period of special scientific interest for PMIP4/past2K. We have revised the sentence and included now references to the PAGES2k Consortium (2019) and the original Wilson et al (2016) N-TREND study. Büntgen et al. (2020) uses only ring-width data and may therefore not always truly be a strong representative record of volcanic forcing on summer temperatures.

----" T*he early 19th century (~1800-1830 CE), at the tail end of the Little Ice Age, marks one of the coldest periods of the last millennium (e.g., Wilson et al, 2016; PAGES2k Consortium, 2019) and is therefore of special interest in the study of inter-decadal climate variability (Jungclaus et al., 2017)."*

**A note on the paragraph starting in line 78: the authors detail changes in their atmospheric component but they only mention in passing that there were also changes in their land component. That is fine, but the authors probably also can be briefer in describing the atmospheric model.**

We have revised the following sentences in the model description to:

*"It consists of four components: the atmospheric general circulation model ECHAM6 (Stevens et al., 2013), the ocean-sea ice model MPIOM (Jungclaus et al., 2013), the land component JSBACH (Reick et*

*al., 2013) and the marine biogeochemistry model HAMOCC (Ilyina et al., 2013). JSBACH is directly coupled to the ECHAM6 model and includes dynamic vegetation, whereas HAMOCC is directly coupled to the MPIOM. ECHAM6 and MPIOM are in turn coupled through the OASIS3-MCT coupler software. In MPI-ESM1.2, ECHAM6.3 is used, which is run with a horizontal resolution in the spectral space of T63 (~200 km) and with 47 vertical levels up to 0.01 hPa with 13 model levels above 100 hPa. In ECHAM6.3 aerosol microphysical processes are not included. The radiative forcing of the volcanic aerosol is prescribed by monthly and zonal mean optical parameters which are generated with the EVA forcing generator, see section 2.1.2. The MPIOM, which is run in its GR15 configuration with a nominal resolution of 1.5°around the equator and 40 vertical levels, has remained largely unchanged with respect to the CMIP5 version. Several revisions with respect to the MPI-ESM CMIP5 version have been made for the atmospheric model including a new representation of radiation transfer, land physics and biogeochemistry components and the ocean carbon cycle. A detailed description of all updates is given in Mauritsen et al. (2019)."*

**Figures: some of the Figures have a strangely low resolution (on my screen). While this will be caught by the technical editing at Copernicus anyway, I wanted to mention it.**

Thank you very much for this hint, we will carefully go over the figures again and will revise them if necessary.

***Data: EEIC. I have to admit I am not really up to date and may express my ignorance but how does the EEIC data differ or improve on the most recent ICOADS data?***

The EEIC data are included in the ICOADS reconstruction, and appear to represent the primary data source for the 1790-1830 period (Freeman et al., 2016), especially over the ocean areas presented in our study. Our analysis could well have used the ICOADS gridded data, and we expect the results would not depend on the choice of data set.

**Data: Stations. I was thinking there should exist tropical stations for the period of interest. Do they have too little temporal coverage, or were they not of interest?**

There is only one tropical station with a long series: Chennai ,India (13.5°N, 80°E) but this station has an unfortunate gap right after the eruption, so we consider NH extratropical station only.

**At line 284 I was wondering whether a very short comparison between reconstructed and observed results would be of value there.**

We have revised Fig.2 by including the reconstructed spatial distribution of NH summer 1811 to cover a three year post volcanic period. The NH summer 1816 panel is therefore shifted to the supplementary material. As we compare reconstruction, station data and model simulation already in the discussion session, we have only slightly revised the text here.

*"The cooling after the 1809 eruption is not so pronounced as after the Mt. Tambora eruption in 1815. In general the station data support the spatial distribution of the reconstructed near surface temperature anomalies derived from tree-ring data. They show a local minimum over Northern, Eastern and Southern Europe in NH summer 1810, which does not appear over Western and Central Europe and New England. The warm anomalies in the order of 2°C, which are found in summer 1811 over Eastern Europe are not however captured by the N-TREND spatial reconstruction although some slight warming is seen in the data over East Poland, Belarus and the Baltic States."*

[Figure]

**F***igure 2*: *Observed and reconstructed temperature anomalies around the 1809 volcanic eruption. **a)** Reconstructed tropical (30°N–30°S, 34°E-70°W) sea surface temperature (TROP, D'Arrigo et al., 2009), measured tropical marine surface air temperatures from ship logs (EEIC, Brohan et al., 2012) and warm pool data (WPOOL, D'Arrigo et al., 2006). **b)** NH summer land temperatures from four tree-ring based reconstructions (Wilson et al,, 2016 (N-TREND (N)), Anchukaitis et al., 2017 (N-TREND (S)), Guillet et al., 2017 (NVOLC), Schneider et al., 2015 (SCH15)). **c)** Monthly mean NH winter and summer temperature anomalies (°C) from 53 station data averaged over different European regions (Central Europe (CEUR: 46.1-52.5°N, 6-17.8°E), Eastern Europe (EEUR: 47-57°N, 18-32°E), Northern Europe (NEUR: 55-66°N, 10 -31° E), Southern Europe (38-46°N, 7-13.5°E), Western Europe (WEUR: 48.5-56°N, 6°W- 6°O) and New England (NENG: 41-44°N, 73- 69°W). **d-f)** Mean surface temperature anomalies (°C) for boreal summers of 1809 (d), 1810 (e) and 1811 (f) in NH TR data N-TREND (S) (Anchukaitis et al, 2017). Pink dots in panel d illustrate the location of the tree-ring proxies used in the N-TREND reconstruction. All anomalies are with respect to the 1800-1808 climatology.*

**Index comparisons: Line 347ff. I fail to see the relevance of the circulation results for the manuscript. They feel unrelated to the rest of the manuscript. They are again referred to in the discussions but to me it also remains unclear there why they are relevant for the argument of the manuscript. The point in the discussion does not depend on the analysis, does it? This is not necessarily a problem but skipping the relevant parts may make the manuscript more concise.**

The analysis was motivated by the idea that the variable response patterns identified between the ensembles but also within each ensemble reflected a lack of robust imprint of volcanic forcing on modes of large-scale climatic variability. Our analysis overall seems to confirm this, especially for the most relevant NAO and PNA. We will tone down the results on the climatic indices to better reflect the specific aim of this analysis. Specifically, we have moved appendix 1 to the supplement and revised the text as follows:

lines 233-237 of the original manuscript are moved to the supplement

the paragraph in lines 347-367 of the original manuscript is moved to the supplement, and replaced with a shortened version along the following lines: *"The substantial differences found in the*

*post-eruption evolution of continental and subcontinental climates that can be produced by internal climate variability and forcing structure reflect substantial differences in the post-eruption evolution. Specifically, post-eruption anomalies of selected dominant modes of large-scale atmospheric circulation in the Northern Hemisphere and the tropics, including the Pacific/North American pattern, the North Atlantic Oscillation, the North Pacific Index and the Southern Oscillation, yield a spread of responses within individual ensembles that is often as large as the range of pre-eruption variability. Further, response distributions generated by different forcings in some cases do not overlap (see Supplementary Figure S1).* "

**Discussions of central Asia: The authors give good reasons for potentially weak reconstructions in Asia and particularly central Asia. Has there been a general evaluation of how well the MPI-ESM performs in these regions?**

Unfortunately, there exist no extensive and detailed model evaluation of the MPI-ESM1.2 over central Asia. Recently, an assessment of 30 CMIP6 models in simulating precipitation over arid Central Asia was published (Guo et al., 2021) but it does not include the MPI-ESM1.2. There exists some PMIP mid Holocene studies which concentrate more on Central Asia but they are more general and partially related to previous versions of the model (e.g. Zheng et al., 2013). However, a comparison of the climatological mean state of near-surface air temperature with ERA-Interim shows that the MPI-ESM-LR model overestimates the mean temperature over the Central Asian highlands about a few degrees (1-5 ℃), slightly underestimates it between 50-65°N and overestimates it again over East Siberia, north of the Arctic circle (Müller et al., 2018; Figure 7a). So we cannot rule out that part of the weak correlation is related to the model biases, which we already briefly mentioned in our conclusion: *"….In contrast, the reconstructed temperature pattern over Asia is not produced by any model simulation, forced or control. Possible explanations for this model-data disagreement include deficiencies and uncertainties regarding both tools. In particular, the dendrochronological network remains ….*"

To emphasize this point a bit more, we have added now a sentence about the model performance in the description of Figure 12:

"T*his is most likely related to a substantial data quality issue as the tree-ring data, especially for central Asia, is solely based on TRW data (Wilson et al., 2016). However, we can also not rule out the possibility of a model biases as the climatological mean state of near-surface air temperature in the MPI-ESM-LR over Central Asia deviates from ERA-Interim about a few degrees (Müller et al., 2018).*"

**TR: I invite the authors to skip the abbreviation TR. "Tree ring" is not too long to be spelled out everywhere.**

We use tree-ring now throughout the text.

**NA: similarly to TR, is it really necessary to abbreviate North America?**

Certainly it is not necessary, we spell North America now out.

**There are a small number of grammar/spelling/typos etc that I assume are artefacts from tracking changes in a document. I only mention Line 655: proxi -> proxy**

We went carefully over the text again and corrected grammar and spelling typos.

*Is a "comparative assessment" a comparison?*

It is similar but it is not the same: a "comparative assessment" includes an evaluating component. We agree that it might be more appropriate to use the more general term "comparison" and have revised the text accordingly.

**I wonder whether it makes sense to spell out S in line 638.**

After the comments from the editor we have skipped this sentence.

**References:**

Anchukaitis, K.J., R. Wilson, K. Briffa, U. Büntgen, E.R. Cook, R.D. D'Arrigo, N. Davi, J. Esper, D. Frank, B. Gunnarson, G. Hegerl, S. Helama, S. Klesse, P.J. Krusic, H. Linderholm, V. Myglan, T. J. Osborn, Z. Peng, M. Rydval, L. Schneider, A. Schurer, G. Wiles and E. Zorita, Last millennium Northern Hemisphere summer temperatures from tree rings: Part II: spatially resolved reconstructions, Quaternary Science Reviews, 163, 1-22, doi: 10.1016/j.quascirev.2017.02.020, 2017.

Brohan, P., Allan, R., Freeman, E., Wheeler, D., Wilkinson, C., and Williamson, F.: Constraining the temperature history of the past millennium using early instrumental observations, Clim. Past, 8, 1551–1563, https://doi.org/10.5194/cp-8-1551-2012, 2012.

Büntgen, Ulf, et al. "Prominent role of volcanism in Common Era climate variability and human history." Dendrochronologia 64 (2020): 125757.

Chenoweth, M. Two major volcanic cooling episodes derived from global marine air temperature, AD 1807–1827 https://doi.org/10.1029/2000GL012648, 2001.

D'Arrigo, R., Wilson, R., Palmer, J., Krusic, P., Curtis, A., Sakulich, J., Bijaksana, S., Zulaikah, S., Ngkoimani, L. O., and Tudhope, A.: The reconstructed Indonesian warm pool sea surface temper-atures from tree rings and corals: Linkages to Asian monsoon drought and El Niño–Southern Oscillation, Paleoceanography,21, PA3005, doi:10.1029/2005pa001256, 2006.

D'Arrigo, R., Wilson R., and Tudhope, A.: The impact of volcanic forcing on tropical temperatures during the past four centuries, Nat. Geosci., 2, 51–56, doi:10.1038/ngeo393, 2009.

Freeman, E., et al.: ICOADS Release 3.0: a major update to the historical marine climate record, Int. J. Climatol., 37(5), 2211–2232, doi:10.1002/joc.4775, 2017.

Guillet, S., et al.: Climateresponse to the Samalas volcanic eruption in 1257 revealed by proxy records, Nature geoscience, 10, 123–128, 2017.

Guo, Hao, et al. Assessment of CMIP6 in simulating precipitation over arid Central Asia. Atmospheric Research, 2021, 252. Jg., S. 105451.

Ilyina, T., Six, K. D., Segschneider, J., Maier-Reimer, E., Li, H., and Nunez-Riboni, I.: Global ocean biogeochemistry model HAMOCC: Model architecture and performance as component of the MPI-Earth system model in different CMIP5 experimental realizations, Journal of Adva--nces in Modeling Earth Systems, 5, 287-315, 10.1029/2012ms000178, 2013.

Jungclaus, J. H., Fischer, N., Haak, H., Lohmann, K.,Marotzke, J., Matei, D., Mikolajewicz, U., Notz, D., and von Storch, J. S.: Characteristics of the ocean simulations in the Max Planck Institute Ocean Model (MPIOM) the ocean component of the MPI-Earth system model: Mpiom CMIP5 Ocean Simulations, J. Adv. Model. Earth Sy., 5, 422–446, https://doi.org/10.1002/jame.20023, 2013.

Jungclaus, J. H. et al: The PMIP4 contribution to CMIP6 – Part 3: The last millennium, scientific objective, and experimental design for the PMIP4 past1000 simulations, Geosci. Model Dev., 10, 4005–4033, https://doi.org/10.5194/gmd-10-4005-2017, 2017.

Mauritsen, T., et al..: A higher-resolution version of the Max Planck Institute Earth System Model (MPI-ESM 1.2-HR), J. Adv. Model. Earth Sy., 10, 1383–1413, https://doi.org/10.1029/2017MS001217, 2018.

PAGES2K "Consistent multi-decadal variability in global temperature reconstructions and simulations over the Common Era." Nature geoscience 12.8 (2019): 643.

Reick, C. H., Raddatz, T., Brovkin, V., and Gayler, V.: Representation of natural and anthropogenic land cover change in MPI-ESM, Journal of Advances in Modeling Earth Systems, 5, 459-482, 10.1002/jame.20022, 2013.

Stevens, B., et al. .: Atmospheric component of the MPI-M Earth System Model: ECHAM6: ECHAM6, J. Adv. Model. Earth Syst., 5(2), 146–172, doi:10.1002/jame.20015, 2013.

Schneider, L., Smerdon, J. E., Büntgen, U., Wilson, R. J. S.,Myglan, V. S., Kirdyanov, A. V., and Esper, J.: Revising mid-latitude summer temperatures back to A.D. 600 based on a latewood density network, Geophys. Res. Lett., 42, GL063956, https://doi.org/10.1002/2015gl063956, 2015.

Toohey, M., Stevens, B., Schmidt, H., and Timmreck, C.: Easy Volcanic Aerosol (EVA v1.0): an idealized forcing generator for climate simulations, Geosci. Model Dev., 9, 4049 – 4070, doi:10.5194/gmd-9-4049-2016, 2016.

Wilson, R., et al.: Last millennium northern hemisphere summer temperatures from tree rings: Part I: The long term context, Quat. Sci. Rev., 134, 1–18, doi:10.1016/j.quascirev.2015.12.005, 2016.

Zanchettin, D., Khodri, M., Timmreck, C., Toohey, M., Schmidt, A., Gerber, E. P. et al.: The Model Intercomparison Project on the climatic response to Volcanic forcing (VolMIP): experimental design and forcing input data for CMIP6, Geosci. Model Dev., 9, 2701 – 2719, doi:10.5194/gmd-9-2701-2016, 2016.

Zheng, W., Wu, B., He, J., and Yu, Y.: The East Asian Summer Monsoon at mid-Holocene: results from PMIP3 simulations, Clim. Past, 9, 453–466, https://doi.org/10.5194/cp-9-453-2013, 2013.